# A SOLVABLE MODEL OF INFERENCE-TIME SCALING

## ABSTRACT

Recent developments in large language models have shown advantages in reallocating a notable share of computational resource from training time to inference time. However, the principles behind inference time scaling are not well understood. In this paper, we introduce an analytically tractable model of inference-time scaling: Bayesian linear regression with a reward-weighted sampler. We study this problem in the high-dimensional regime, where the deterministic equivalents dictate a closed-form expression for the posterior predictive mean and variance. We analyze the generalization error when training data are sampled from a teacher model. We draw $k$ inference-time samples and select via softmax at a temperature applied to a quadratic reward. When the reward is not too different from the teacher, the generalization error decreases monotonically with increasing inference time samples $k$. However, the specific reward that optimizes inference-time selection generally differs from the teacher. In contrast, substantial reward misspecification induces a finite optimal $k$ beyond which more sampling can increase the generalization error, consistent with recent empirical observations. Furthermore, for fixed $k$, there exists an optimal sampling temperature. In the "best-of-$k$" limit with the teacher as reward, we prove that the generalization error decays as $\Theta(1/k^2)$ and determine the leading coefficient via extreme value theory. These formulas delineate domains where scaling inference-time computation is provably preferable to collecting more data. Finally, we demonstrate that when task difficulty increases, the previously mentioned advantage of inference-time compute degrades.

## 1 INTRODUCTION

Across tasks, allowing models to 'think longer' at inference by sampling multiple candidates, re-ranking with a reward, or aggregating votes consistently improves precision and reliability (Wang et al., 2023; Wu et al., 2024; Snell et al., 2025). Best-of-$k$ (choose the highest-reward sample) and majority voting (choose the consensus) have become standard inference-time tools (Brown et al., 2024; Schaeffer et al., 2025b; Chen et al., 2024a). In parallel, scaling training compute via larger models and more data also drives dramatic gains (Hestness et al., 2017; Kaplan et al., 2020; Hoffmann et al., 2022). Modern systems, therefore, have to balance two resources, training-time and test-time compute, and practitioners use both.

Despite widespread adoption, key questions for inference-time computation lack crisp answers. Which reward model should we use for inference? What is the appropriate inference-time sampler, e. g., temperature settings? How large should $k$ be and when do more samples stop helping? Most importantly, how should we allocate a fixed compute budget between training and inference to minimize generalization error? We lack a simple, solvable model that gives us intuitions and actionable prescriptions.

To fill this gap, we propose a minimal and analytically tractable setting: Bayesian regression based on a teacher-student setting with a controlled reward and a temperature-dependent inference-time sampler, in which best-of-$k$ appears as limit. We theoretically study the generalization error $\delta$ as a function of the size of the training data set $n$, the dimension of the data $d$, the number of inference time samples $k$, the sampling temperature $T$ and the reward parameter $\mathbf{w}_R$ and demonstrate various optimality conditions on these parameters.

Our contributions are the following:

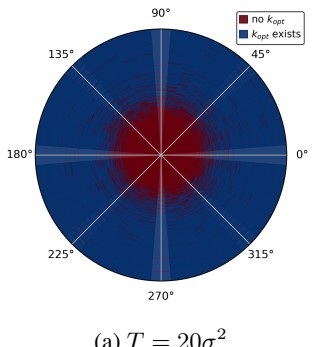 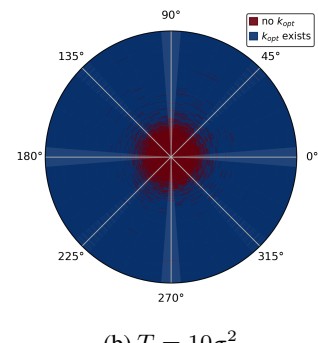

(a) $T = 20\sigma^2$          (b) $T = 10\sigma^2$

Figure 1: In the plot the radial distance is the magnitude $c$ of the vector $\mathbf{w}_R - \mathbf{w}_T$ and the polar variable is the angle $\theta$ between $\mathbf{w}_R - \mathbf{w}_T$ and $\mathbf{w}_T$. We see that as temperature $T$ decreases the domain where generalization error $\delta$ decreases monotonically with increase in inference-time samples $k$ shrinks. We have chosen $S = 1, \sigma = 10^{-4}, \gamma = 10^{-3}, d = 2, n = 10^4$ and sampled teacher weight $\mathbf{w}_T = (c\cos\theta_T, c\sin\theta_T) \sim \mathcal{N}(0, 2^2\mathbf{I})$. We have parameterized the reward weight as follows: $\mathbf{w}_R = \mathbf{w}_T + (c\cos(\theta_T + \theta), c\sin(\theta_T + \theta)), \theta \in [0, 2\pi)$. See section 2 for details of notation and conventions.

- We propose a solvable model for inference-time scaling: Bayesian regression where the ground truth is given by the teacher model $y = \mathbf{w}_T \cdot \mathbf{x}/\sqrt{d}$ and the reward function is quadratic $r(y, \mathbf{x}) = -(y - \mathbf{w}_R \cdot \mathbf{x}/\sqrt{d})^2$. We generate $k$ samples at inference and choose using a softmax at temperature $T$ over the reward. We present a formula for the generalization error $\delta$ in the proportional limit: $d \to \infty, n \to \infty$ with $\alpha = d/n$ fixed.

- We derive a series expansion for $\delta$ around large $T$, making explicit its dependence on $n$, $k$, and the alignment between $\mathbf{w}_R$ and $\mathbf{w}_T$. The analytical expansion shows a sharp dependence on reward quality: when $\mathbf{w}_R$ is sufficiently close to $\mathbf{w}_T$, i.e., small $\|\mathbf{w}_R - \mathbf{w}_T\|/\|\mathbf{w}_T\|$, increasing $k$ monotonically decreases $\delta$, and the reward $\mathbf{w}_R$ that optimizes inference-time selection generally *differs* from the data-generating teacher $\mathbf{w}_T$. In contrast, when $\mathbf{w}_R$ is poorly aligned, $\delta$ is non-monotone in $k$, yielding an *optimal* finite $k$ (see Figure 1), echoing phenomena observed empirically in large language models (Snell et al., 2025). Furthermore we show that at fixed $k$, there exists an *optimal* temperature $T$ for the rewarding process. Similar yet distinct observation has been made in large language models (Du et al., 2025) for sampling temperature of the model itself. We emphasize that our result is about the sampling temperature of the rewarding process - different from the sampling temperature of the language model.

- We have generated inference time samples from Meta-Llama-3-8B-Instruct on openai/gsm8k validation dataset (prompt included 8 chain of thought demonstrations). For each question and response pair we used Mistral-7B-Instruct-v0.3 to generate a reward score. Finally, we observed the existence of an optimal $k, T$ similar to the theoretical predictions above.

- For $T = 0$, using extreme value theory, we analytically prove that the expectation value of $\delta$ scales as $\Theta(1/k^2)$ at large $k$ when we have access to the teacher, i.e, $\mathbf{w}_R = \mathbf{w}_T$. Based on our theoretical analysis in best-of-$k$ limit, we quantify the parametric region where scaling inference-time compute is more *beneficial* compared to training compute. Finally we note that when task difficulty increases, the previously mentioned advantage of inference-time compute degrades.

Now we turn to survey the ideas in the literature related to our work.

## 1.1 RELATED WORKS

**Method of deterministic equivalence.** In the context of linear regression (Krogh & Hertz, 1992; Dicker, 2016; Dobriban & Wager, 2018; Nakkiran, 2019; Advani et al., 2020; Hastie et al., 2022),

kernel regression (Sollich, 1998; Sollich & Halees, 2002; Bordelon et al., 2020; Canatar et al., 2021; Spigler et al., 2020; Simon et al., 2023; Loureiro et al., 2021), and random feature models (Hastie et al., 2022; Louart et al., 2018; Mei & Montanari, 2022; Adlam & Pennington, 2020; d'Ascoli et al., 2020; d'Ascoli et al., 2020; Loureiro et al., 2021; Bahri et al., 2022; Zavatone-Veth & Pehlevan, 2023a; Dhifallah & Lu, 2020; Hu & Lu, 2022; Maloney et al., 2022; Bach, 2024) method of deterministic equivalence (Voiculescu et al., 1992; Zee, 1996) has been used extensively for discussions of higher dimensional statistics (Misiakiewicz & Saeed, 2024; Atanasov et al., 2024). These ideas have been used to discuss training time scaling laws in simple models (Spigler et al., 2020; Bordelon et al., 2020; Bahri et al., 2022; Maloney et al., 2022; Simon et al., 2021; Bordelon et al., 2024; Zavatone-Veth & Pehlevan, 2023b; Paquette et al., 2024; Lin et al., 2024; Bordelon et al., 2025). We use this technique to simplify the posterior probability distribution of the Bayesian regression model.

**Inference-time scaling.** A growing body of work investigates how to allocate and exploit inference-time compute to improve predictive performance, with empirical gains reported across tasks and domains based on majority voting (Chen et al., 2024a; Snell et al., 2025; Setlur et al., 2025; Arora & Zanette, 2025; Wu et al., 2024; Liu et al., 2025; Du et al., 2025) or a best-of-$k$ strategy (Wang et al., 2023; Yao et al., 2023a; Brown et al., 2024; Levi, 2024; Schaeffer et al., 2025a; Huang et al., 2025; Chen et al., 2024b; Du et al., 2025; Chen et al., 2025). These procedures are often paired with reasoning-oriented prompting and structured search that expand the candidate set before selection (e.g., chain-of-thought and tree-of-thoughts) (Wei et al., 2022; Yao et al., 2023b). The work of Chen et al. (2024a) presented a theoretical model for majority voting in the premise of classification problems. More close to our work is the theoretical model of Levi (2024) on best-of-$k$ strategy. For a given trained model, both these works explain some of the empirically observed patterns at inference. We study similar questions for the regression model and discuss trade-off between training and inference time compute taking into account the quality of the reward and the sampling process.

## 2 PROBLEM SETUP: BAYESIAN REGRESSION WITH REWARD-WEIGHTED SAMPLING

We start by introducing our solvable model.

### 2.1 TRAINING METHOD - PRIOR AND POSTERIOR DISTRIBUTION

We study a supervised regression setting with a linear teacher model that maps inputs to outputs and then add observation noise. Throughout, let $\mathbf{x} \in \mathbb{R}^d$ denote an input vector drawn from a zero-mean Gaussian with covariance $\boldsymbol{\Sigma}$, written $\mathbf{x} \sim \mathcal{N}(0, \boldsymbol{\Sigma})$. We assume $\text{Tr}(\boldsymbol{\Sigma}) = \Theta_d(d)$ so that the total feature variance scales linearly with dimension; a canonical case is $\boldsymbol{\Sigma} = \mathbf{I}$. The teacher parameter $\mathbf{w}_T$ is taken to have norm $\|\mathbf{w}_T\|^2 = d$, and the output is given by:

$$y = \mathbf{w}_T \cdot \frac{\mathbf{x}}{\sqrt{d}} + \eta, \quad \eta \sim \mathcal{N}(0, \sigma^2). \tag{1}$$

Given a training set $\mathcal{D} = \left\{ (\mathbf{x}^i, y^i)_{i=1}^n \right\}$ sampled i.i.d. from the teacher, we adopt a Bayesian linear regression perspective with an isotropic Gaussian prior on the weights, $\mathcal{N}(0, \gamma^2 \mathbf{I})$. Bayes' rule yields the posterior distribution over weights:

$$p(\mathbf{w}|\mathcal{D}) = \frac{p(\mathcal{D}|\mathbf{w})p(\mathbf{w})}{p(\mathcal{D})}. \tag{2}$$

Predictions for a new test input $\mathbf{x}$ are obtained by marginalizing the likelihood under this posterior, producing the *posterior predictive* distribution:

$$p(y|\mathbf{x}, \mathcal{D}) = \int d\mathbf{w} \, p(\mathbf{w}|\mathcal{D})p(y|\mathbf{x}, \mathbf{w}). \tag{3}$$

Next we state the standard result that makes predictive distribution explicit.

**Lemma 1.** *Analytical formula for the posterior predictive is given by*

$$p(y|\mathbf{x}, \mathcal{D}) = \mathcal{N}\left(\boldsymbol{\mu} \cdot \frac{\mathbf{x}}{\sqrt{d}}, \frac{\mathbf{x}}{\sqrt{d}}^{\top} \boldsymbol{\Omega} \frac{\mathbf{x}}{\sqrt{d}} + \sigma^2\right) \tag{4}$$

$$\boldsymbol{\mu} = \frac{1}{\sigma^2}\boldsymbol{\Omega}\sum_{i=1}^{n} y^i \frac{\mathbf{x}^i}{\sqrt{d}}, \quad \boldsymbol{\Omega}^{-1} = \frac{1}{\sigma^2}\sum_{i=1}^{n}\frac{\mathbf{x}^i}{\sqrt{d}}\frac{\mathbf{x}^i}{\sqrt{d}}^{\top} + \frac{1}{\gamma^2}\mathbf{I} \tag{5}$$

*Proof.* Consult Bishop (2013) for the proof. □

## 2.2 INFERENCE-TIME SAMPLING AND THE REWARD MODEL

Suppose that we have a reward model that evaluates our predictions, $r(y, \mathbf{x})$. We will use this to generate an output with the following procedure:

---
Reward-Weighted Sampling

---
**Require:** input $\mathbf{x}$, posterior predictive $p(y \mid \mathbf{x}, \mathcal{D})$, reward $r$, temperature $T$, number of samples $k$
1: **for** $i \leftarrow 1$ to $k$ **do**
2:     sample $y_i \sim p(y \mid \mathbf{x}, \mathcal{D})$
3:     $w_i \leftarrow \exp\big(r(y_i, \mathbf{x})/T\big)$
4: $q_i \leftarrow w_i / \sum_{j=1}^{k} w_j \quad (i = 1, \ldots, k)$
5: Draw $I \sim \text{Categorical}(q_1, \ldots, q_k)$
6: **return** $y_{\text{out}} \leftarrow y_I$

---

For simplicity, we will assume a reward given by

$$r(y, \mathbf{x}) = -\left(y - \mathbf{w}_R \cdot \frac{\mathbf{x}}{\sqrt{d}}\right)^2 \tag{6}$$

Note that $\mathbf{w}_R \neq \mathbf{w}_T$ in general.

In this paper, we are interested in computing the generalization error of this model defined by

$$\delta = \mathbb{E}_{\mathbf{x}} \mathbb{E}_{y_1, \ldots, y_k}\left[\frac{\sum_{i=1}^{k}\left(y_i - \mu_T(\mathbf{x})\right)^2 e^{-\left(y_i - \mu_R(\mathbf{x})\right)^2/T}}{\sum_{j=1}^{k} e^{-\left(y_j - \mu_R(\mathbf{x})\right)^2/T}}\right] \tag{7}$$

Here we use the notation $\mu_T(\mathbf{x}) := \mathbf{w}_T \cdot \frac{\mathbf{x}}{\sqrt{d}}, \mu_R(\mathbf{x}) := \mathbf{w}_R \cdot \frac{\mathbf{x}}{\sqrt{d}}$.

## 3 ANALYSIS OF THE GENERALIZATION ERROR

In this section we analyze the high-dimensional behavior of the Bayesian regression model introduced above, with a particular focus on how inference-time sampling and the reward model shape the error. In the high dimensional setting, deterministic-equivalents simplify the predictive mean and variance by $m$ and $\Sigma = s^2$ as follows:

**Theorem 1.** *In the limit of $d, n \to \infty$, with $\alpha = d/n < 1$ fixed, the generalization error is given by*

$$\delta = \mathbb{E}_{\mathbf{x} \sim \mathcal{N}(0, \boldsymbol{\Sigma})} \mathbb{E}_{y_i \sim \mathcal{N}(m(\mathbf{x}), s(\mathbf{x})^2), i=1,2,\ldots,k}\left[\frac{\sum_{i=1}^{k}\left(y_i - \mu_T(\mathbf{x})\right)^2 e^{-\left(y_i - \mu_R(\mathbf{x})\right)^2/T}}{\sum_{j=1}^{k} e^{-\left(y_j - \mu_R(\mathbf{x})\right)^2/T}}\right], \tag{8}$$

*when noise scale is sufficiently small. Here the posterior predictive has a deterministic mean $m(\mathbf{x})$ and variance $\Sigma(\mathbf{x}) = s(\mathbf{x})^2$ as follows*

$$m(\mathbf{x}) = \frac{\mathbf{x}}{\sqrt{d}}^{\top} \mathbf{A}_R \mathbf{w}_T, \quad s(\mathbf{x})^2 = \sigma^2 + \gamma^2 \frac{\mathbf{x}}{\sqrt{d}}^{\top} \mathbf{B}_R \frac{\mathbf{x}}{\sqrt{d}}. \tag{9}$$

*The matrices $\mathbf{A}_R, \mathbf{B}_R$ are given by*

$$\mathbf{A}_R := \mathbf{\Sigma}(\mathbf{\Sigma} + R\mathbf{I})^{-1}, \qquad \mathbf{B}_R := R(\mathbf{\Sigma} + R\mathbf{I})^{-1} = \mathbf{I} - \mathbf{A}_R. \tag{10}$$

*and the renormalized ridge $R$ is given by*

$$\hat{R} = R\left(1 - \alpha\, m_{\mathbf{\Sigma}}(R)\right) = \frac{\sigma^2}{\gamma^2}\alpha, \qquad m_{\mathbf{\Sigma}}(R) := \frac{1}{d}Tr\left[\mathbf{\Sigma}(\mathbf{\Sigma} + RI)^{-1}\right]. \tag{11}$$

*Proof.* See appendix B for more details. $\qquad\square$

In this paper we will focus on the simple setup where $\mathbf{\Sigma} = S^2\mathbf{I}$. In this case we can explicitly solve for the renormalized ridge as follows

$$R = \frac{1}{2}S^2\left(\frac{d}{n} + \frac{\hat{R}}{S^2} - 1 + \left(\left(1 - \frac{d}{n} - \frac{\hat{R}}{S^2}\right)^2 + \frac{4\hat{R}}{S^2}\right)^{\frac{1}{2}}\right) \tag{12}$$

In addition in this case the matrices $\mathbf{A}_R, \mathbf{B}_R$ are proportional to identity matrix

$$\mathbf{A}_R := \frac{S^2}{R + S^2}\mathbf{I}, \qquad \mathbf{B}_R := \frac{R}{R + S^2}\mathbf{I}. \tag{13}$$

These expressions are going to be useful in the later sections to have close form expression of generalization error $\delta$ in various parameter domains of interest.

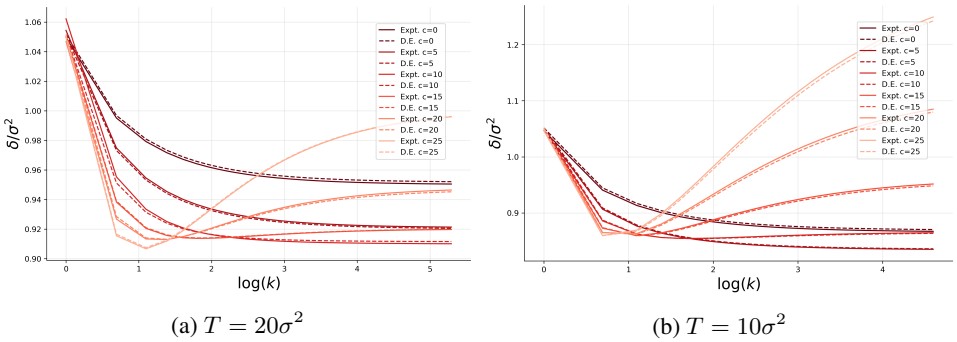

(a) $T = 20\sigma^2$ $\qquad\qquad\qquad\qquad\qquad$ (b) $T = 10\sigma^2$

Figure 2: In the plot we have chosen $S = 1, \sigma = 10^{-4}, \gamma = 10^{-3}, n = 10^4, d = 10^1$ and sampled teacher weight $\mathbf{w}_T \sim \mathcal{N}(0, 2^2\mathbf{I})$. We have parameterized the reward weight as follows: $\mathbf{w}_R = (1 + cR/(R + S^2))\mathbf{w}_T$. Solid and dashed lines correspond to the experimental results and the formula in Theorem 1 respectively.

Before we get into the details of further theoretical analysis, we summarize the empirical findings in Figure 2 and compare with results of Theorem 1. When the reward is sufficiently accurate—formally, when $\|\mathbf{w}_R - \mathbf{w}_T\|/\|\mathbf{w}_T\|$ is small—the generalization error $\delta$ decreases *monotonically* with the number of inference-time samples $k$. We denote by $\mathcal{R}$ (highlighted in red in Figure 1) the region of reward weights exhibiting this monotonic behavior. Notably, within $\mathcal{R}$ the teacher reward is *not* optimal for fixed $(k, T)$; i.e., $\mathbf{w}_R = \mathbf{w}_T$ does not minimize $\delta$ (see Figure 2a). Comparing figures 2a–2b, we note that the set $\mathcal{R}$ shrinks as the temperature $T$ decreases (see also Figure 1). Outside this set, in its complement $\mathcal{R}^{\complement}$ (blue in Figure 1), Figure 2 shows that $\delta$ becomes *non-monotonic* in $k$, with a finite $k$ beyond which increasing $k$ worsens error. Consequently, for a fixed $\mathbf{w}_R$, lowering $T$ can induce a transition from $\mathcal{R}$ to $\mathcal{R}^{\complement}$; equivalently, at fixed $k$ there may exist an *optimal* temperature $T$ that minimizes $\delta$ (confirmed in Figure 3c).

We now provide a theoretical account of these phenomena. Specifically, we analyze $\delta$ in two complementary regimes of the reward temperature: (i) a high-temperature (weak-reward) expansion, where the selection reweighting is perturbative, and (ii) a low-temperature ("best-of-$k$") regime, where selection concentrates on high-reward samples and extreme-value effects dominate. The next two results formalize these regimes.

**Theorem 2** (High-$T$ expansion). *For $T \gg s(\mathbf{x})^2$ the expectation value of the error can be organized as a perturbative series as follows*

$$\delta \;=\; \mathbb{E}_{\mathbf{x}}\Big[\Delta_T(\mathbf{x})^2 + s^2(\mathbf{x}) + \Sigma_{l=1}^3 (-1)^l \frac{C_l(\mathbf{x})}{t(\mathbf{x})^l} \prod_{i=1}^l \Big(1 - \tfrac{i}{k}\Big) + \mathcal{O}\Big(t(\mathbf{x})^{-4}\Big)\Big] \tag{14}$$

*Where we have defined*

$$C_l(\mathbf{x}) = 2\,\Delta_T(\mathbf{x})\Delta_R(\mathbf{x}) + s^2(\mathbf{x}) + (l-1)\Delta_R(\mathbf{x})^2 \tag{15}$$

$$\Delta_T(\mathbf{x}) := m(\mathbf{x}) - \mu_T(\mathbf{x}), \quad \Delta_R(\mathbf{x}) := m(\mathbf{x}) - \mu_R(\mathbf{x}), \quad t(\mathbf{x}) = \frac{T}{2s(\mathbf{x})^2} \tag{16}$$

*and all other quantities are as in Theorem 1.*

*Proof.* Let $z$ denote the partition function over $k$ i.i.d. draws from $p(y|\mathbf{x}, \mathcal{D})$ with quadratic reward; expand $\mathbb{E}\log z$ around $\mathbb{E}z$ via a controlled the cumulant expansion for $t \gg 1$. The $1/t$, $1/t^2$ and $1/t^3$ terms produce the $C_1(\mathbf{x}), C_2(\mathbf{x})$ and $C_3(\mathbf{x})$ structure; substituting the deterministic equivalents for $m, \Sigma$ converts it to the form mentioned in the Theorem. See Appendix C for the details. □

In the limit of an ample amount of data $d/n \to 0$ with a flat prior $\sigma/\gamma \to 0$, the temperature scale is controlled by $s^2 \approx \sigma^2$. Hence the Theorem above is valid in the high-temperature limit in that sense.

**Theorem 3** (Low-$T$ best-of-$k$ sampling). *When we have access to the exact teacher weight $\mathbf{w}_R = \mathbf{w}_T = \mathbf{w}$, the leading order result for $T \to 0$ followed by $k \to \infty$ is given by*

$$\delta \;=\; \frac{\pi}{k^2}\,\mathbb{E}_{\mathbf{x}}\Big[s^2(\mathbf{x})\,\exp\Big(\frac{\Delta_T(\mathbf{x})^2}{s^2(\mathbf{x})}\Big)\Big] \tag{17}$$

*All the quantities are as in Theorem 1 and Theorem 2.*

*Proof.* At $T = 0$, the softmax reduces to a minimum of chi-squared random variables. This is governed by the Weibull distribution at large $k$ according to extreme value theory. Finally, substituting the deterministic equivalents for $m, \Sigma$ and evaluating the expectation value gives the generalization error mentioned in the Theorem. See Appendix 11 for details. □

Note that the low-temperature scaling of $\delta$ with the number of inference time samples $k$ is independent of the amount of the amount of training data.

These theoretical results are compared with the experiment in Figure 4.

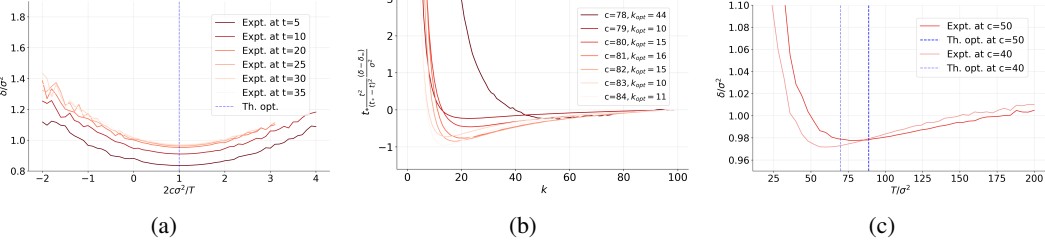

|  (a)  |  (b)  |  (c)  |

Figure 3: In the plot we have chosen $S = 1, \sigma = 10^{-4}, \gamma = 10^{-3}, n = 10^4, d = 10$ and used the following parameterization $\mathbf{w}_R = (1 + cR/(R + S^2))\mathbf{w}_T$ and sampled $\mathbf{w}_T \sim \mathcal{N}(0, 2^2\mathbf{I})$. (a) This plot shows dependence of $\delta$ on $c$ for various values of $T$. In these graphs we have kept $k = 50$ fixed. We see that $\delta$ is minimized at $c \approx T/(2\sigma^2)$ as expected from lemma 2. (b) We plot the scaled value of $\delta - \delta_\infty$, $\delta_\infty \approx \delta_{k=100}$ as a function of $k$ for various values of $c$. This shows existence of an optimal value of $k$ - theoretical prediction for it is denoted as $k_{opt}$ as given in lemma 3. In this plot we have chosen a fixed temperature $T = 200\sigma^2$ (c) This plot shows existence of an optimal value of temperature $T$ at fixed $k = 50$. For the optimal value, we find reasonable agreement with the theoretical prediction in in lemma 4.

### 3.1 Does the teacher represent the optimal reward?

Figure 3a compares the generalization error achieved when the reward weight equals the teacher ($\mathbf{w}_R = \mathbf{w}_T$) versus when it differs, across different values of $k$ and $T$. The plot reveals a consistent pattern: when $\mathbf{w}_R$ is close to $\mathbf{w}_T$, the error is *lower* when the reward weight is *slightly shifted* away from the teacher. This additional shift required for the optimal reward grows systematically with the temperature scale $T$. This fact is explained by the following lemma:

**Lemma 2.** *There exists an optimal reward weight that differs from the teacher weight by the following formula*

$$\mathbf{w}_R(\mathbf{x}) = \mathbf{w}_T + \left( \frac{k}{k-2} t(\mathbf{x}) \right) \mathbf{B}_R \, \mathbf{w}_T \tag{18}$$

*This formula is valid in the domain stated in Theorem 2 as long as*

$$\frac{\|\mathbf{w}_R(\mathbf{x}) - \mathbf{w}_T\|}{\|\mathbf{w}_T\|} \ll 1 \tag{19}$$

*Proof.* This is obtained by setting the first derivative of $\delta$, given in Theorem 2, with respect to $\mathbf{w}_R$ to zero. $\qquad \square$

The lemma also quantifies the empirically observed fact that as $T$ increases, the optimal $\mathbf{w}_R$ moves proportionally away from $\mathbf{w}_T$.

### 3.2 Is there an optimal value for inference-time samples?

Figure 3b shows that when the reward weight $\mathbf{w}_R$ is sufficiently misaligned from the teacher $\mathbf{w}_T$, the test error as a function of the number of inference samples $k$ is *non-monotonic*: it first decreases (benefiting from a better draw among $k$ candidates) and then *increases* beyond an optimal value of $k$.

We can get some more insight into this behavior by the following lemma:

**Lemma 3.** *There exists an optimal value of inference samples $k$, when Theorem 2 is valid and*

$$t < \frac{3C_2(\mathbf{x})}{C_1(\mathbf{x})} \equiv t_*, \quad C_1(\mathbf{x}) > 0, \quad C_2(\mathbf{x}) > 0 \tag{20}$$

*In this case, as we increase $k$ the error decreases until when we reach*

$$k \approx \frac{2t_*}{t_* - t} \tag{21}$$

*and far beyond this, increase in $k$ increases the error. For larger values of $t$, increase in $k$ always decreases error.*

*Proof.* This is obtained by setting the first derivative of $\delta$, given in Theorem 2, with respect to $k$ to zero. $\qquad \square$

Note that above lemma only applies when $t \gg 1$ and $t < 3C_2/C_1$. But when $\mathbf{w}_R$ is sufficiently close to $\mathbf{w}_T$, $C_2/C_1 \sim 1$. When $\mathbf{w}_R$ is sufficiently close to $\mathbf{w}_T$, increase in $k$ decreases the error $\delta$ since in this case above lemma does not apply. Whereas when $\mathbf{w}_R$ is sufficiently far from $\mathbf{w}_T$ ($C_2/C_1$ becomes larger since $\Delta_R$ grows), above lemma shows existence of an optimal value for $k$.

### 3.3 Is there an optimal temperature?

Figure 3c examines generalization $\delta$ error as a function of temperature $T$ at fixed number of inference time samples $k$. Empirically, the generalization error exhibits a clear local minimum around a critical value of $T$, rather than decreasing or increasing monotonically. Interpreting this through the high-$T$ expansion, the minimum corresponds to a particular balance between the first- and second-order correction terms governed by $C_1$ and $C_2$. Temperature $T$ controls how sharply the selection favors high-reward samples among the $k$ candidates. At very high temperatures, selection is nearly

uniform and the benefits of the reward model are muted; at very low temperatures, selection becomes too aggressive and can over-amplify any mismatch between $\mathbf{w}_R$ and $\mathbf{w}_T$, increasing error. The optimal temperature thus trades off these effects. It scales linearly with $\Sigma$ and grows with misspecification via $C_2/C_1$. The location of the optimal temperature for given $k, \mathbf{w}_R$ is determined from the theoretical result below:

**Lemma 4.** *For a given number of inference-time samples $k > 2$, training dataset size $n$ and reward weight $\mathbf{w}_R$, there exists an optimal temperature for the rewarding process*

$$t(\mathbf{x}) = 2\left(1 - \frac{2}{k}\right)\frac{C_2(\mathbf{x})}{C_1(\mathbf{x})}, \quad C_1(\mathbf{x}) > 0, \quad C_2(\mathbf{x}) > 0 \tag{22}$$

*This formula is valid in the domain stated in Theorem 2.*

*Proof.* This is obtained by setting the first derivative of $\delta$, given in Theorem 2, with respect to $t$ to zero. □

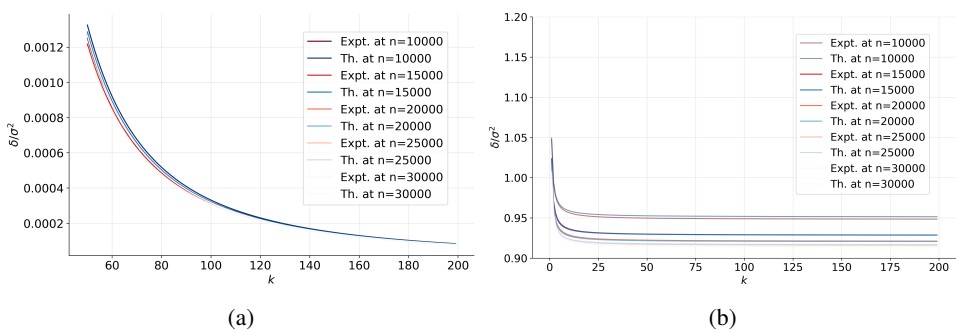

(a)                                           (b)

Figure 4: In the plot we have chosen $S = 1, \sigma = 10^{-4}, \gamma = 10^{-3}, d = 10$. (a) We plot the theoretical (given in Theorem 3) and experimental value of $\delta$ at $T = 0$ and find good agreement for $\mathbf{w}_R = \mathbf{w}_T \sim \mathcal{N}(0, 2^2\mathbf{I})$. We see that in this domain scaling $n$ higher is less useful compared to scaling $k$. (b) We plot the theoretical (given in Theorem 2) and experimental value of $\delta$ and find good agreement for $\mathbf{w}_R = \mathbf{w}_T \sim \mathcal{N}(0, 2^2\mathbf{I})$. We see that in this domain scaling $n$ higher is more useful compared to scaling $k$.

### 3.4 WHEN WE HAVE ACCESS TO THE TEACHER AS THE REWARD MODEL, HOW DOES GENERALIZATION ERROR DECAY FOR THE BEST OF $k$ SAMPLING - EXPONENTIALLY OR AS A POWER LAW?

The low-$T$ result in Theorem 3 shows an inverse–quadratic $k^{-2}$ decay of the error when the reward matches the teacher ($\mathbf{w}_R = \mathbf{w}_T$). Here we sharpen that statement by identifying a concrete, practically relevant parameter domain in which the leading-order constant in front of $k^{-2}$ can be written in closed form. Explicit formula clarifies how dimensionality, sample size, noise, and prior scale combine in the low-temperature limit.

**Lemma 5.** *As a refinement of Theorem 3, consider the parameter regime*

$$\frac{\gamma^2}{d}Tr(\mathbf{B}_R\Sigma) \ll \sigma^2.$$

*In the low-temperature limit $T \to 0$ followed by $k \to \infty$, the leading-order generalization error for $\mathbf{w}_R = \mathbf{w}_T = \mathbf{w}$ is given by*

$$\delta = \frac{\pi\sigma^2}{k^2}\frac{1}{\sqrt{1 - \frac{2}{\sigma^2 d}\mathbf{u}^\top\Sigma\mathbf{u}}}, \quad \mathbf{u} := \mathbf{B}_R\mathbf{w}$$

*Proof.* In this domain the $\mathbf{x}$-dependence of $s^2(\mathbf{x})$ is small relative to $\sigma^2$ and this allows us to reliably set $s^2(\mathbf{x}) \approx \sigma^2$ in Theorem 3 to evaluate the expectation value. □

In the flat prior limit, i.e, $\gamma^2 \gg \sigma^2$, this regime corresponds to ample amount of data per dimension, i.e., $n \gg d$. For the isotropic sample covariance $\mathbf{\Sigma} = S^2\mathbf{I}$ that we are analyzing in this paper, we have $\mathbf{u} = \frac{R}{R+S^2}\mathbf{w}$. In the limit of flat prior with ample amount of data, this further simplifies to $\mathbf{u} \approx (1/S^2)(\sigma^2/\gamma^2)(d/n)\mathbf{w}$. The lemma above shows as task difficulty increases, i.e, $\sigma$ gets larger keeping other parameters fixed, generalization error $\delta$ and even the scaled generalization error $\delta/\sigma^2$ increases.

### 3.5 What is the trade-off between training and inference-time compute?

In practice we often face a budget allocation decision: should additional compute be spent on *training* (e.g., acquiring/processing more samples $n$) or on *inference-time* (e.g., drawing more candidates $k$ and selecting via the reward)? When the reward is well aligned with the teacher and we operate in the low-temperature regime, best-of-$k$ style selection can substantially reduce error with relatively modest inference cost. The question is how this compares with the addition of more data.

**Lemma 6.** *Given access to exact teacher weight, it is beneficial to scale inference-time compute over adding more training samples in the following regime: consider $T \to 0$ followed by $k \to \infty$ with*

$$\frac{\gamma^2}{d}Tr(\mathbf{B}_R\mathbf{\Sigma}) \ll \sigma^2, \quad R \ll \sigma^2 \tag{23}$$

*That is within this domain,*

$$\frac{\partial \log \delta}{\partial \log k} = -2, \qquad \frac{\partial \log \delta}{\partial \log n} = -\frac{\alpha\partial_\alpha\left(\mathbf{u}^\top\mathbf{\Sigma}\mathbf{u}\right)}{\sigma^2 d - 2\mathbf{u}^\top\mathbf{\Sigma}\mathbf{u}}, \qquad \left|\frac{\partial \log \delta}{\partial \log k}\right| \gg \left|\frac{\partial \log \delta}{\partial \log n}\right| \tag{24}$$

*Proof.* See Appendix E for details. $\square$

In the flat prior, i.e, $\gamma^2 \gg \sigma^2$, ample data limit, i.e., $n \gg d$, the second condition in the lemma quantifies prior quality - roughly speaking it dictates that when the prior $\gamma$ is broad enough, inference time compute is beneficial over training time compute. For the isotropic sample covariance $\mathbf{\Sigma} = S^2\mathbf{I}$, putting back explicit formula for $\mathbf{u} \approx R/S^2\mathbf{w} \approx (1/S^2)(\sigma^2/\gamma^2)(d/n)\mathbf{w}$ shows that

$$\frac{\partial \log \delta}{\partial \log n} = -2\frac{\frac{\mathbf{w}\cdot\mathbf{w}}{d}\frac{1}{S^2}\frac{d^2\sigma^2}{n^2\gamma^4}}{1 - 2\frac{\mathbf{w}\cdot\mathbf{w}}{d}\frac{1}{S^2}\frac{d^2\sigma^2}{n^2\gamma^4}} \tag{25}$$

In this case, under an even weaker condition $R^2 < \sigma^2$ already we see that scaling inference time is beneficial over scaling training compute. If $\sigma$ increases keeping other parameters held fixed, the magnitude of the derivative of $\log \delta$ w.r.t. $\log n$ increases. Hence as the task becomes more difficult the advantage of inference time scaling degrades. The same statement holds true if $\gamma$ decreases while other parameters are held fixed.

We empirically validate these results on advantages of inference-time scaling over training compute in Figure 5b. It is clear from Figure 5b that fractional increase in $k$ decreases generalization error $\delta$ more compared to the same fractional increase in $n$ within the domain of parameters considered in the plot. However, inference-time scaling is not always advantageous over increasing training compute - we explain this in Figure 4b. We conclude that when we have access to a good quality reward model and the task is easy enough, the addition of inference time compute is beneficial over additional training compute.

## 4 Qualitative agreement with large language model reasoning

In this section we discuss implications of our results for inference time scaling of large language models.

In the linear model we have observed that the when the reward is not close to the teacher model there exists an optimal value of inference time samples. This fact has been observed in large language models in (Snell et al., 2025; Chen et al., 2024a). We present similar results in figure 5.

Our second observation is that there exists an optimal temperature for reward sampling. To the best of our knowledge this is a new observation and in this section we present experiments supporting

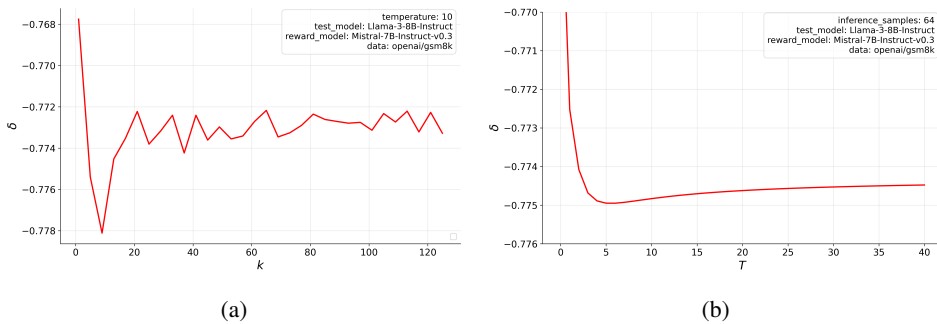

(a)  (b)

Figure 5: In the plot we have generated inference time samples from Meta-Llama-3-8B-Instruct on openai/gsm8k validation dataset (prompt included 8 chain of thought demonstrations). For each question and response pair we used Mistral-7B-Instruct-v0.3 to generate a reward score. For generalization error we used the definition in equation 26.

it. For a given question $\mathbf{x}$ we generate $k$ responses $y_i, i = 1, 2, \ldots, k$ from the large language model under study and use a judge language model to assign a reward $r(y_i, \mathbf{x})$ to $i$ th response. The generalization error is defined by

$$\delta = -\mathbb{E}_{\mathbf{x}} \sum_{i=1}^{k} \frac{e^{\frac{r(y_i, \mathbf{x})}{T}}}{\sum_{j=1}^{k} e^{\frac{r(y_j, \mathbf{x})}{T}}} \, v(y_i, \mathbf{x}) \tag{26}$$

Here $v(y_i, \mathbf{x}) \in \{0, 1\}$ is 1 when the response is correct and 0 otherwise. When $T = 0$ this reduces to the best of $k$ rewarding process. In figure 5 we present experimental results confirming that there exists an optimal value of $T$.

In the linear setting, we show that the reward model that optimizes performance need not coincide with the teacher model. Leveraging properties of the trained predictor, specifically, that its mean prediction approaches the teacher's value from below, we derive the optimal reward model. By analogy, in large language models one can employ an auxiliary classifier that learns the model's systematic weaknesses and selects which queries should be scored by the reward model; such classifier-guided reward shaping can further improve performance. A comprehensive study of these directions is left to future work.

We have also discussed trade-off between training and inference time scaling, analyzing it carefully in large language models is an important open question.

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

## A REVIEW OF EXTREME VALUE STATISTICS

### A.1 LIMIT LAWS FOR MAXIMA

Here we note some of the useful results in extreme value theory from de Haan & Ferreira (2007).

**Theorem 4** (Fisher–Tippett–Gnedenko). *Let $X_1, X_2, \dots$ be i.i.d. non-degenerate random variables with distribution function $F$, i.e, $F(x) = \mathbb{P}(X \leq x)$, and let $M_n = \max\{X_1, \dots, X_n\}$. If there exist normalising constants $a_n > 0$ and $b_n \in \mathbb{R}$ and a non-degenerate distribution function $H$ such that*

$$\mathbb{P}\left(\frac{M_n - b_n}{a_n} \leq x\right) \xrightarrow[n \to \infty]{} H(x) \qquad (x \in \mathbb{R} \text{ continuity points of } H),$$

*then $H$ must be (up to affine change of variable) one of the three* extreme value distributions*:*

$$\text{Fréchet } (\alpha > 0): \quad \Phi_\alpha(x) = \begin{cases} 0, & x \leq 0, \\ \exp\{-x^{-\alpha}\}, & x > 0, \end{cases}$$

$$\text{Weibull } (\alpha > 0): \quad \Psi_\alpha(x) = \begin{cases} \exp\{-(-x)^\alpha\}, & x \leq 0, \\ 1, & x > 0, \end{cases}$$

$$\text{Gumbel}: \quad \Lambda(x) = \exp\{-e^{-x}\}, \qquad\qquad\qquad x \in \mathbb{R}.$$

### A.2 MAXIMUM DOMAINS OF ATTRACTION AND NORMING CONSTANTS

**Definition 5** (Maximum domain of attraction). *We say $F$ belongs to the maximum domain of attraction of $H$ (write $F \in \mathrm{MDA}(H)$) if there exist constants $a_n > 0, b_n \in \mathbb{R}$ such that*

$$\lim_{n \to \infty} F^n(a_n x + b_n) = H(x) \qquad (x \in \mathbb{R} \text{ continuity points of } H).$$

**Proposition 1** (Characterisation via exceedance rates). *Let $H$ be a (standard) extreme value distribution. Then $F \in \mathrm{MDA}(H)$ with norming constants $a_n > 0, b_n \in \mathbb{R}$ if and only if*

$$\lim_{n \to \infty} n\big(1 - F(a_n x + b_n)\big) = -\ln H(x) \qquad (x \in \mathbb{R}).$$

For later convenience we define right endpoint $x_F := \sup\{x : F(x) < 1\}$, complementary distribution function $\bar{F}(x) := 1 - F(x)$ and quantile function $F^\leftarrow(t) = \inf\{x \in \mathbb{R} : F(x) \geq t\}$.

#### A.2.1 THE MAXIMUM DOMAIN OF ATTRACTION OF THE FRÉCHET DISTRIBUTION

**Theorem 6** (MDA of Fréchet). *Let $F$ have a finite right endpoint $x_F = \infty$ and and assume there exists $z < x_F$ such that $F$ is differentiable in $(z, x_F)$ The following statements are equivalent:*

*(i) $F$ satisfies* von Mises condition*, i.e.,*

$$\lim_{x \to x_F-} \frac{x F'(x)}{1 - F(x)} = \alpha > 0 \tag{27}$$

*(ii) $F \in \mathrm{MDA}(\Phi_\alpha)$ with a possible choice of norming constants*

$$b_n = 0, \qquad a_n = F^\leftarrow\left(1 - \frac{1}{n}\right),$$

### A.2.2 THE MAXIMUM DOMAIN OF ATTRACTION OF THE WEIBULL DISTRIBUTION

**Theorem 7** (MDA of Weibull). *Let $F$ have a finite right endpoint $x_F < \infty$ and and assume there exists $z < x_F$ such that $F$ is differentiable in $(z, x_F)$ The following statements are equivalent:*

*(i) $F$ satisfies* von Mises condition, *i.e.,*

$$\lim_{x \to x_F-} \frac{(x_F - x)\, F'(x)}{1 - F(x)} = \alpha > 0 \tag{28}$$

*(ii) $F \in \mathrm{MDA}(\Psi_\alpha)$ with a possible choice of norming constants*

$$b_n = x_F, \qquad a_n = x_F - F^{\leftarrow}\left(1 - \frac{1}{n}\right),$$

### A.2.3 THE MAXIMUM DOMAIN OF ATTRACTION OF THE GUMBEL DISTRIBUTION

**Theorem 8** (MDA of Gumbel). *Let $F$ be a distribution with right endpoint $x_F \leq \infty$, and assume there exists $z < x_F$ such that $F$ is at least twice differentiable in $(z, x_F)$. Define auxiliary function $a(x) = \bar{F}(x)/F'(x)$. The following statements are equivalent:*

*(i) $F$ is a von Mises function, i.e.,*

$$\lim_{x \to x_F-} a'(x) = 0$$

*(ii) $F \in MDA(\Lambda)$ with a possible choice of norming constants*

$$b_n = F^{\leftarrow}(1 - 1/n), \qquad a_n = a(b_n)$$

## B  PROOF OF THEOREM 1

**Theorem 9.** *In the limit of $d, n \to \infty$, with $\alpha = d/n < 1$ fixed, the generalization error is given by*

$$\delta = \mathbb{E}_{\mathbf{x} \sim \mathcal{N}(0, \mathbf{\Sigma})}\, \mathbb{E}_{y_i \sim \mathcal{N}(m(\mathbf{x}), s(\mathbf{x})^2), i=1,2,\ldots,k} \left[ \frac{\sum_{i=1}^{k} \left(y_i - \mu_T(\mathbf{x})\right)^2 e^{-\left(y_i - \mu_R(\mathbf{x})\right)^2/T}}{\sum_{j=1}^{k} e^{-\left(y_j - \mu_R(\mathbf{x})\right)^2/T}} \right], \tag{29}$$

*when noise scale is sufficiently small. Here the posterior predictive has a deterministic mean $m(\mathbf{x})$ and variance $\Sigma(\mathbf{x}) = s(\mathbf{x})^2$ as follows*

$$m(\mathbf{x}) = \frac{\mathbf{x}^\top}{\sqrt{d}} \mathbf{A}_R \mathbf{w}_T, \quad s(\mathbf{x})^2 = \sigma^2 + \gamma^2 \frac{\mathbf{x}^\top}{\sqrt{d}} \mathbf{B}_R \frac{\mathbf{x}}{\sqrt{d}}. \tag{30}$$

*The matrices $\mathbf{A}_R, \mathbf{B}_R$ are given by*

$$\mathbf{A}_R := \mathbf{\Sigma}(\mathbf{\Sigma} + R\mathbf{I})^{-1}, \qquad \mathbf{B}_R := R(\mathbf{\Sigma} + R\mathbf{I})^{-1} = \mathbf{I} - \mathbf{A}_R. \tag{31}$$

*and the renormalized ridge $R$ is given by*

$$\hat{R} = R\left(1 - \alpha\, m_{\mathbf{\Sigma}}(R)\right) = \frac{\sigma^2}{\gamma^2}\alpha, \qquad m_{\mathbf{\Sigma}}(R) := \frac{1}{d} Tr\left[\mathbf{\Sigma}(\mathbf{\Sigma} + R I)^{-1}\right]. \tag{32}$$

*Proof.* Let the empirical covariance be $\widehat{\mathbf{\Sigma}} := \frac{1}{n}\sum_{i=1}^{n} \mathbf{x}^i {\mathbf{x}^i}^\top$, then

$$\mathbf{\Omega} = \sigma^2 \alpha \left(\widehat{\mathbf{\Sigma}} + \hat{R}\,\mathbf{I}\right)^{-1}, \qquad \hat{R} := \frac{\sigma^2}{\gamma^2}\alpha. \tag{33}$$

The mean of the predictive is given by

$$m(\mathbf{x}) = \boldsymbol{\mu}^\top \frac{\mathbf{x}}{\sqrt{d}} = \frac{1}{\sigma^2} \frac{\mathbf{x}^\top}{\sqrt{d}} \mathbf{\Omega} \sum_{i=1}^{n} y^i \frac{\mathbf{x}^i}{\sqrt{d}} = \frac{1}{\sigma^2} \frac{\mathbf{x}^\top}{\sqrt{d}} \mathbf{\Omega} \left(\sum_{i=1}^{n} \frac{\mathbf{x}^i}{\sqrt{d}} \frac{{\mathbf{x}^i}^\top}{\sqrt{d}}\right) \mathbf{w}_T + \frac{1}{\sigma^2} \frac{\mathbf{x}^\top}{\sqrt{d}} \mathbf{\Omega} \sum_{i=1}^{n} \eta^i \frac{\mathbf{x}^i}{\sqrt{d}}$$

$$= \frac{\mathbf{x}^\top}{\sqrt{d}} \left(\mathbf{I} - \frac{1}{\gamma^2}\mathbf{\Omega}\right) \mathbf{w}_T + \frac{1}{\sigma^2} \frac{\mathbf{x}^\top}{\sqrt{d}} \mathbf{\Omega} \sum_{i=1}^{n} \eta^i \frac{\mathbf{x}^i}{\sqrt{d}}, \tag{34}$$

Now we use $\mathbf{I} - (1/\gamma^2)\mathbf{\Omega} = \mathbf{I} - \hat{R}\left(\widehat{\mathbf{\Sigma}} + \hat{R}\mathbf{I}\right)^{-1} = \widehat{\mathbf{\Sigma}}\left(\widehat{\mathbf{\Sigma}} + \hat{R}\mathbf{I}\right)^{-1}$ to get

$$m(\mathbf{x}) = \left\langle \widehat{\mathbf{\Sigma}}\left(\widehat{\mathbf{\Sigma}} + \hat{R}\mathbf{I}\right)^{-1}\mathbf{w}_T, \frac{\mathbf{x}}{\sqrt{d}} \right\rangle + \frac{1}{\sigma^2}\frac{\mathbf{x}}{\sqrt{d}}^\top \mathbf{\Omega} \sum_{i=1}^{n} \eta^i \frac{\mathbf{x}^i}{\sqrt{d}}, \tag{35}$$

Conditioned test data $X = [\mathbf{x}^1, \ldots, \mathbf{x}^n]$ and the test $\mathbf{x}$, the vector $\sum_{i=1}^{n} \eta^i \frac{\mathbf{x}^i}{\sqrt{d}}$ has zero mean with conditional covariance $\sigma^2 \sum_{i=1}^{n} \frac{\mathbf{x}^i}{\sqrt{d}} \frac{\mathbf{x}^i}{\sqrt{d}}^\top$. Hence when $\sigma$ is sufficiently small and $\alpha < 1$ we can ignore this contribution.

The variance of the predictive can be simplified to

$$s^2(\mathbf{x}) = \frac{\mathbf{x}}{\sqrt{d}}^\top \mathbf{\Omega} \frac{\mathbf{x}}{\sqrt{d}} + \sigma^2 = \frac{\mathbf{x}}{\sqrt{d}}^\top \left[\sigma^2\alpha(\widehat{\mathbf{\Sigma}} + \hat{R}\mathbf{I})^{-1}\right]\frac{\mathbf{x}}{\sqrt{d}} + \sigma^2$$
$$= \gamma^2 \frac{\mathbf{x}}{\sqrt{d}}^\top \left[\hat{R}(\widehat{\mathbf{\Sigma}} + \hat{R}\mathbf{I})^{-1}\right]\frac{\mathbf{x}}{\sqrt{d}} + \sigma^2, \tag{36}$$

There exists a $R > 0$ as defined in the Theorem such that, for any vectors $u, v$ with bounded norms,

$$u^\top\left[\widehat{\mathbf{\Sigma}}(\widehat{\mathbf{\Sigma}} + \hat{R}\mathbf{I})^{-1}\right]v \xrightarrow{p} u^\top\left[\mathbf{\Sigma}(\mathbf{\Sigma} + R\mathbf{I})^{-1}\right]v = u^\top\mathbf{A}_R v, \tag{37}$$
$$u^\top\left[\hat{R}\left(\widehat{\mathbf{\Sigma}} + \hat{R}\mathbf{I}\right)^{-1}\right]v \xrightarrow{p} u^\top\left[R(\mathbf{\Sigma} + R\mathbf{I})^{-1}\right]v = u^\top\mathbf{B}_R v. \tag{38}$$

Applying equation 37 to equation 35, yields

$$m(\mathbf{x}) \xrightarrow{p} \frac{\mathbf{x}}{\sqrt{d}}^\top \mathbf{A}_R\mathbf{w}_T$$

Applying equation 38 to equation 36 with $u = v = \frac{\mathbf{x}}{\sqrt{d}}$ yields

$$s^2(\mathbf{x}) \xrightarrow{p} \sigma^2 + \gamma^2 \frac{\mathbf{x}}{\sqrt{d}}^\top \mathbf{B}_R \frac{\mathbf{x}}{\sqrt{d}}.$$

$\square$

## C  PROOF OF THEOREM 2

**Theorem 10.** *For $T \gg s(\mathbf{x})^2$ the expectation value of the error can be organized as a perturbative series as follows*

$$\delta = \mathbb{E}_\mathbf{x}\left[\Delta_T(\mathbf{x})^2 + s^2(\mathbf{x}) + \Sigma_{l=1}^3(-1)^l\frac{C_l(\mathbf{x})}{t(\mathbf{x})^l}\prod_{i=1}^{l}\left(1 - \tfrac{i}{k}\right) + \mathcal{O}\left(t(\mathbf{x})^{-4}\right)\right] \tag{39}$$

*Where we have defined*

$$C_l(\mathbf{x}) = 2\,\Delta_T(\mathbf{x})\Delta_R(\mathbf{x}) + s^2(\mathbf{x}) + (l-1)\Delta_R(\mathbf{x})^2 \tag{40}$$

$$\Delta_T(\mathbf{x}) := m(\mathbf{x}) - \mu_T(\mathbf{x}), \quad \Delta_R(\mathbf{x}) := m(\mathbf{x}) - \mu_R(\mathbf{x}), \quad t(\mathbf{x}) = \frac{T}{2s(\mathbf{x})^2} \tag{41}$$

*and all other quantities are as in Theorem 1.*

*Proof.* Consider a random variable $x$ and we are interested in concentration properties of the function $f(x) = \log x$. Here we will discuss a controlled approximation technique to evaluate the expectation value of $f$. Simplest approach is to Taylor expand $f$ around the expectation value of $x$ denoted by $\bar{x}$

$$f(x) = \log \bar{x} + \frac{1}{\bar{x}}(x - \bar{x}) - \frac{1}{2\bar{x}^2}(x - \bar{x})^2 + \frac{1}{3\bar{x}^3}(x - \bar{x})^3 + \mathcal{O}\left(\frac{(x - \bar{x})^4}{\bar{x}^4}\right) \tag{42}$$

Taking expectation value of both sides of the equation above we get the following expression

$$\mathbb{E}(\log x) = \log \mathbb{E}(x) - \frac{\mathbb{E}(x^2) - \mathbb{E}(x)^2}{2\mathbb{E}(x)^2} + \frac{\mathbb{E}(x^3) - 3\mathbb{E}(x^2)\mathbb{E}(x) + 2\mathbb{E}(x)^3}{3\mathbb{E}(x)^3} + \mathcal{O}\left(\frac{\mathbb{E}((x - \mathbb{E}(x))^4)}{\mathbb{E}(x)^4}\right) \tag{43}$$

This approximation scheme is only useful when higher order corrections are relatively small. Next we use this to derive the result stated in the Theorem above.

It will be convenient to define partition function density given by

$$z(\mathbf{J}, \mathbf{p}) = \frac{1}{k} \sum_{i=1}^{k} e^{-\frac{1}{T} E_{\mathbf{w}_R}(p_i) - J_i E_{\mathbf{w}_T}(p_i)}, \quad E_{\mathbf{w}}(p) = (p - \mathbf{w} \cdot \frac{\mathbf{x}}{\sqrt{d}})^2 \tag{44}$$

The expectation value of $z(\mathbf{J}, \mathbf{p})$ when $p_i$ is sampled from the following distribution

$$p_i \sim \mathcal{N}(m, \Sigma = s^2), \quad i = 1, 2, \ldots, k \tag{45}$$

gives disorder averaged, over $k$ samples, partition function, with an additional chemical potential $\mathbf{J}$, of free particles at temperature $T$. The error given in (7) can be expressed in terms of this quantity as follows

$$\delta = \mathbb{E}_{\mathbf{p}}\left(\frac{\sum_{i=1}^{k} E_{\mathbf{w}_T}(p_i) e^{-\frac{1}{T} E_{\mathbf{w}_R}(p_i) - J_i E_{\mathbf{w}_T}(p_i)}}{\sum_{i=1}^{k} e^{-\frac{1}{T} E_{\mathbf{w}_R}(p_i) - J_i E_{\mathbf{w}_T}(p_i)}}\right)$$

$$= -\mathbb{E}_{\mathbf{p}} \sum_{i} \partial_{J_i}(\log z(\mathbf{J}, \mathbf{p}))|_{\mathbf{J}=0}$$

$$\approx -\sum_{i} \partial_{J_i}\left(\log \mathbb{E}_{\mathbf{p}}(z(\mathbf{J}, \mathbf{p})) - \frac{\mathbb{E}_{\mathbf{p}}(z(\mathbf{J}, \mathbf{p})^2) - \mathbb{E}_{\mathbf{p}}(z(\mathbf{J}, \mathbf{p}))^2}{2\mathbb{E}_{\mathbf{p}}(z(\mathbf{J}, \mathbf{p}))^2}\right.$$

$$\left.\left. + \frac{\mathbb{E}_{\mathbf{p}}(z(\mathbf{J}, \mathbf{p})^3) - 3\mathbb{E}_{\mathbf{p}}(z(\mathbf{J}, \mathbf{p})^2)\mathbb{E}_{\mathbf{p}}(z(\mathbf{J}, \mathbf{p}) + 2\mathbb{E}_{\mathbf{p}}(z(\mathbf{J}, \mathbf{p}))^3}{3\mathbb{E}_{\mathbf{p}}(z(\mathbf{J}, \mathbf{p}))^3}\right)\right|_{\mathbf{J}=0} \tag{46}$$

To go from the second to the third line we have made an approximation, we will present the domain of validity of the approximation shortly. To this end we compute,

$$\mathbb{E}_{\mathbf{p}}(z(\mathbf{J}, \mathbf{p})) = \frac{1}{k} \sum_{i=1}^{k} \mathbb{E}_{p}(e^{-\frac{1}{T} E_{\mathbf{w}_R}(p) - J_i E_{\mathbf{w}_T}(p)}) \tag{47}$$

The required expectation values can be expressed in terms of the following function

$$h(m_1, m_2, m_3; s_1^2, s_2^2, s_3^2) = \frac{\exp\left[-\dfrac{\displaystyle\sum_{1 \le i \ne j \ne k \le 3}\left(\frac{1}{2}m_i^2(s_j^2 + s_k^2) - m_i m_j s_k^2\right)}{2 \prod_{i=1}^{3} s_i^2 \sum_{i=1}^{3} \frac{1}{s_i^2}}\right]}{2\pi\sqrt{\prod_{i=1}^{3} s_i^2 \sum_{i=1}^{3} \frac{1}{s_i^2}}}. \tag{48}$$

The moments of $z(\mathbf{J}, \mathbf{p})$ are given by

$$\mathbb{E}_{\mathbf{p}}[z(\mathbf{J}, \mathbf{p})] = \frac{1}{k} \sum_{i=1}^{k} \pi\sqrt{\frac{T}{J_i}} h\left(m, \frac{\mathbf{w}_R \cdot \mathbf{x}}{\sqrt{d}}, \frac{\mathbf{w}_T \cdot \mathbf{x}}{\sqrt{d}}; s^2, \frac{T}{2}, \frac{1}{2J_i}\right), \tag{49}$$

$$\mathbb{E}_{\mathbf{p}}[z(\mathbf{J}, \mathbf{p})^2] = \frac{1}{k^2}\left[\sum_{\substack{i,j=1 \\ i \ne j}}^{k} \pi\sqrt{\frac{T}{J_i}} h\left(m, \frac{\mathbf{w}_R \cdot \mathbf{x}}{\sqrt{d}}, \frac{\mathbf{w}_T \cdot \mathbf{x}}{\sqrt{d}}; s^2, \frac{T}{2}, \frac{1}{2J_i}\right) \pi\sqrt{\frac{T}{J_j}} h\left(m, \frac{\mathbf{w}_R \cdot \mathbf{x}}{\sqrt{d}}, \frac{\mathbf{w}_T \cdot \mathbf{x}}{\sqrt{d}}; s^2, \frac{T}{2}, \frac{1}{2J_j}\right)\right.$$

$$\left. + \sum_{i=1}^{k} \pi\sqrt{\frac{T}{4J_i}} h\left(m, \frac{\mathbf{w}_R \cdot \mathbf{x}}{\sqrt{d}}, \frac{\mathbf{w}_T \cdot \mathbf{x}}{\sqrt{d}}; s^2, \frac{T}{4}, \frac{1}{4J_i}\right)\right], \tag{50}$$

$$\mathbb{E}_{\mathbf{p}}[z(\mathbf{J},\mathbf{p})^3] = \frac{1}{k^3}\left[\sum_{1\le i<j<\ell\le k} 6\,\pi\sqrt{\frac{T}{J_i}}\,h\!\left(m,\frac{\mathbf{w}_R\cdot\mathbf{x}}{\sqrt{d}},\frac{\mathbf{w}_T\cdot\mathbf{x}}{\sqrt{d}};s^2,\frac{T}{2},\frac{1}{2J_i}\right)\right.$$

$$\times\,\pi\sqrt{\frac{T}{J_j}}\,h\!\left(m,\frac{\mathbf{w}_R\cdot\mathbf{x}}{\sqrt{d}},\frac{\mathbf{w}_T\cdot\mathbf{x}}{\sqrt{d}};s^2,\frac{T}{2},\frac{1}{2J_j}\right)\pi\sqrt{\frac{T}{J_\ell}}\,h\!\left(m,\frac{\mathbf{w}_R\cdot\mathbf{x}}{\sqrt{d}},\frac{\mathbf{w}_T\cdot\mathbf{x}}{\sqrt{d}};s^2,\frac{T}{2},\frac{1}{2J_\ell}\right)$$

$$+\sum_{\substack{i,j=1\\i\neq j}}^{k} 3\,\pi\sqrt{\frac{T}{4J_i}}\,h\!\left(m,\frac{\mathbf{w}_R\cdot\mathbf{x}}{\sqrt{d}},\frac{\mathbf{w}_T\cdot\mathbf{x}}{\sqrt{d}};s^2,\frac{T}{4},\frac{1}{4J_i}\right)\pi\sqrt{\frac{T}{J_j}}\,h\!\left(m,\frac{\mathbf{w}_R\cdot\mathbf{x}}{\sqrt{d}},\frac{\mathbf{w}_T\cdot\mathbf{x}}{\sqrt{d}};s^2,\frac{T}{2},\frac{1}{2J_j}\right)$$

$$\left.+\sum_{i=1}^{k}\pi\sqrt{\frac{T}{9J_i}}\,h\!\left(m,\frac{\mathbf{w}_R\cdot\mathbf{x}}{\sqrt{d}},\frac{\mathbf{w}_T\cdot\mathbf{x}}{\sqrt{d}};s^2,\frac{T}{6},\frac{1}{6J_i}\right)\right]. \tag{51}$$

This leads to the following perturbative expansion in terms of $t=\frac{T}{2s^2}$:

$$\delta(\mathbf{x}) = (m-\mathbf{w}_T\cdot\frac{\mathbf{x}}{\sqrt{d}})^2+s^2-\frac{(1-\frac{1}{k})\left((m-\frac{\mathbf{w}_R\cdot\mathbf{x}}{\sqrt{d}})\left(2(m-\frac{\mathbf{w}_T\cdot\mathbf{x}}{\sqrt{d}})\right)+s^2\right)}{t}$$

$$+\frac{(1-\frac{1}{k})(1-\frac{2}{k})\left(\left(m-\frac{\mathbf{w}_R\cdot\mathbf{x}}{\sqrt{d}}\right)\left(2\left(m-\frac{\mathbf{w}_T\cdot\mathbf{x}}{\sqrt{d}}\right)+\left(m-\frac{\mathbf{w}_R\cdot\mathbf{x}}{\sqrt{d}}\right)\right)+s^2\right)}{t^2}+O\!\left(\frac{1}{t^3}\right) \tag{52}$$

Repeating this technique to higher order is straight forward but algebraically tedious. Here we quote the result to one higher order

$$\delta(\mathbf{x}) = (m-\mathbf{w}_T\cdot\frac{\mathbf{x}}{\sqrt{d}})^2+s^2+\Sigma_{l=1}^{3}(-1)^l\left(1-\frac{1}{k}\right)\left(1-\frac{2}{k}\right)\dots\left(1-\frac{l}{k}\right)\frac{1}{t^l}C_l(\mathbf{x})+O\!\left(\frac{1}{t^4}\right) \tag{53}$$

$$C_l(\mathbf{x}) = 2\,(m-\frac{\mathbf{w}_R\cdot\mathbf{x}}{\sqrt{d}})(m-\frac{\mathbf{w}_T\cdot\mathbf{x}}{\sqrt{d}})+s^2(\mathbf{x})+(l-1)(m-\frac{\mathbf{w}_R\cdot\mathbf{x}}{\sqrt{d}})^2 \tag{54}$$

$\square$

# D  PROOF OF THEOREM 3

**Theorem 11** (Low-$T$ best-of-$k$ sampling). *When we have access to the exact teacher weight $\mathbf{w}_R = \mathbf{w}_T = \mathbf{w}$, the leading order result for $T\to 0$ followed by $k\to\infty$ is given by*

$$\delta = \frac{\pi}{k^2}\,\mathbb{E}_{\mathbf{x}}\left[s^2(\mathbf{x})\,\exp\!\left(\frac{\Delta_T(\mathbf{x})^2}{s^2(\mathbf{x})}\right)\right] \tag{55}$$

*All the quantities are as in Theorem 1 and Theorem 2.*

*Proof.* In $T\to 0$ limit it is natural to approximate the softmax by the sample with highest reward, i.e.,

$$\delta(\mathbf{x}) = \mathbb{E}_{y_i\sim\mathcal{N}(m,\Sigma),i=1,\dots,k}\left(\left(y'_k-\mathbf{w}_T\cdot\frac{\mathbf{x}}{\sqrt{d}}\right)^2:y'_k=\arg\min_{y\in(y_1,y_2,..,y_k)}\left(y-\mathbf{w}_R\cdot\frac{\mathbf{x}}{\sqrt{d}}\right)^2\right) \tag{56}$$

First note that the distribution of the penalty is a non-central chi-squared distribution with one degree of freedom:

$$-v\equiv\frac{1}{s^2}\left(y-\mathbf{w}_R\cdot\frac{\mathbf{x}}{\sqrt{d}}\right)^2\sim\chi_1^2(\lambda),\quad\lambda=\frac{(m-\mathbf{w}_R\cdot\frac{\mathbf{x}}{\sqrt{d}})^2}{s^2} \tag{57}$$

We focus on the situation of perfect reward $\mathbf{w}_R=\mathbf{w}_T$ in the high reward regime. In this case,

$$\delta(\mathbf{x}) = s^2\mathbb{E}(-v_{max}),\quad v_{max}=\max_{-v_i\sim\chi_1^2(\lambda)}(v_1,v_2,..,v_k) \tag{58}$$

Since we are looking for the minimum of the chi-squared distributed variables the extreme value statistics is more involved and the final distribution is different compared to the analysis for maximum.

Next we focus on $k \to \infty$ limit for analytical tractability. Probability distribution function $\varphi$ and cumulative distribution function $F$ of $v \leq 0$ is given by

$$\varphi(v) = \frac{e^{\frac{v-\lambda}{2}} \cos\left(\sqrt{\lambda v}\right)}{\sqrt{2\pi}\sqrt{-v}}, \quad F(v) = 1 - \frac{1}{2}\left(\operatorname{erf}\left(\frac{\sqrt{-v} - \sqrt{\lambda}}{\sqrt{2}}\right) + \operatorname{erf}\left(\frac{\sqrt{\lambda} + \sqrt{-v}}{\sqrt{2}}\right)\right)$$

$$(59)$$

Note that as $k \to \infty$ the degenerate distribution concentrates near $v_F = 0$. Given that $v_F$ is finite, the extreme distribution could be either of Weibull or Gumbel type. Next we show that it is not Gumbel type. To see this we calculate the auxiliary function for the Gumbel type using

$$a(v) = \frac{1 - F(v)}{F'(v)}, \quad \lim_{v \to v_F} a'(v) \neq 0 \tag{60}$$

This limit on right ensures it cannot be of Gumbel type. To identify the Weibull distribution

$$\Psi_\alpha(x) = e^{-(-x)^\alpha} \quad \text{for } x \leq 0, \quad 1 \quad \text{otherwise} \tag{61}$$

We calculate

$$\alpha = \lim_{v \to v_F} \frac{(v_F - v)F'(v)}{1 - F(x)} = \frac{1}{2} \tag{62}$$

This ensures the cumulative distribution function of $(v_{max} - d_k)/c_k$ is $\Psi_\alpha$ where

$$d_k = v_F = 0$$
$$c_k = v_F - F^{\leftarrow}\left(1 - \frac{1}{k}\right) = \frac{\pi}{2k^2}e^\lambda \tag{63}$$

The probability density for $-v_{max}/c_k$ is

$$p_{max}(-v_{max}/c_k) = \alpha(-v_{max}/c_k)^{\alpha-1}e^{-(-v_{max}/c_k)^\alpha}, \quad v_{max} \leq 0 \tag{64}$$

This gives the following expression for the error in $t \to 0, k \to \infty$ limit (taken in this order)

$$\delta(\mathbf{x}) = s^2 c_k \mathbb{E}(-v_{max}/c_k) = s^2 c_k \Gamma(1 + 1/\alpha) = \frac{\pi s^2}{k^2} e^{\frac{(m - \mathbf{w} \cdot \frac{\mathbf{x}}{\sqrt{d}})^2}{s^2}} \tag{65}$$

$\square$

## E  PROOF OF LEMMA 6

**Lemma 7.** *Given access to exact teacher weight, it is beneficial to scale inference-time compute over adding more training samples in the following regime: consider $T \to 0$ followed by $k \to \infty$ with*

$$\frac{\gamma^2}{d}Tr(\mathbf{B}_R \boldsymbol{\Sigma}) \ll \sigma^2, \quad R \ll \sigma^2 \tag{66}$$

*That is within this domain,*

$$\frac{\partial \log \delta}{\partial \log k} = -2, \quad \frac{\partial \log \delta}{\partial \log n} = -\frac{\alpha \partial_\alpha \left(\mathbf{u}^\top \boldsymbol{\Sigma} \mathbf{u}\right)}{\sigma^2 d - 2\mathbf{u}^\top \boldsymbol{\Sigma} \mathbf{u}}, \quad \left|\frac{\partial \log \delta}{\partial \log k}\right| \gg \left|\frac{\partial \log \delta}{\partial \log n}\right| \tag{67}$$

*Proof.* Our goal is to put upper bound on the magnitude of the derivative w.r.t. $n$. We proceed systematically by working in the eigen basis of sample variance. By the spectral Theorem for real symmetric matrices, there exists an orthogonal matrix $Q \in O(d)$ and a diagonal $\Lambda = \operatorname{diag}(\lambda_1, \ldots, \lambda_d)$ with $\lambda_i \geq 0$ such that

$$\boldsymbol{\Sigma} = Q\Lambda Q^\top, \quad \mathbf{B}_R = Q \operatorname{diag}\left(\frac{R}{\lambda_i + R}\right)Q^\top.$$

This immediately gives following inequalities

$$m_{\mathbf{\Sigma}}(R) = \frac{1}{d}\mathrm{Tr}\big[\mathbf{\Sigma}(\mathbf{\Sigma} + R\mathbf{I})^{-1}\big] = \frac{1}{d}\sum_{i=1}^{d}\frac{\lambda_i}{\lambda_i + R} < 1 \tag{68}$$

$$m'_{\mathbf{\Sigma}}(R) = -\frac{1}{d}\mathrm{Tr}\big[\mathbf{\Sigma}(\mathbf{\Sigma} + R\mathbf{I})^{-2}\big] = -\frac{1}{d}\sum_{i=1}^{d}\frac{\lambda_i}{(\lambda_i + R)^2} \le 0 \tag{69}$$

We are interested in putting upper bound on the following quantity

$$\alpha\,\partial_\alpha(\boldsymbol{u}^\top\mathbf{\Sigma}\,\boldsymbol{u}) = \alpha\,F'(R)\,\frac{dR}{d\alpha}, \quad F(R) = \boldsymbol{u}^\top\mathbf{\Sigma}\,\boldsymbol{u} = \sum_{i=1}^{d}\frac{R^2\lambda_i}{(\lambda_i + R)^2}\,\tilde{w}_i^{\,2}, \quad \tilde{\boldsymbol{w}} = Q^\top\boldsymbol{w}$$

We will first derive an upper bound on $F'(R)$ and an upper bound on $\frac{dR}{d\alpha}$. To this goal we proceed by noting that $\max_{\lambda \ge 0}\frac{\lambda^2}{(\lambda+R)^3} = \frac{4}{27}\frac{1}{R}$ (attained at $\lambda = 2R$) and we have

$$F'(R) = \sum_{i=1}^{d}\frac{2R\,\lambda_i^2}{(\lambda_i + R)^3}\,\tilde{w}_i^2 \le 2R \cdot \frac{4}{27}\frac{1}{R}\,\|\boldsymbol{w}\|^2 = \frac{8}{27}\,d.$$

Next we focus on the deterministic equivalents equation

$$\hat{R} = R\big(1 - \alpha m_{\mathbf{\Sigma}}(R)\big), \qquad \hat{R} = \frac{\sigma^2\alpha}{\gamma^2}.$$

Differentiate both sides w.r.t. $\alpha$:

$$\Big((1 - \alpha m_{\mathbf{\Sigma}}) - \alpha R m'_{\mathbf{\Sigma}}\Big)\,R' = \frac{\sigma^2}{\gamma^2} + R\,m_{\mathbf{\Sigma}}(R).$$

Using the inequalities in equation (68)

$$R' \;\le\; \frac{\frac{\sigma^2}{\gamma^2} + R\,m_{\mathbf{\Sigma}}(R)}{\hat{R}/R} \;\le\; \frac{\frac{\sigma^2}{\gamma^2} + R}{\hat{R}/R} = \frac{\sigma^2}{\gamma^2}\frac{R}{\hat{R}} + \frac{R^2}{\hat{R}}.$$

This gives us the desired upper bound

$$\alpha\,\partial_\alpha(\boldsymbol{u}^\top\mathbf{\Sigma}\,\boldsymbol{u}) = \alpha\,F'(R)\,\frac{dR}{d\alpha} \;\le\; \frac{8}{27}\,d\Big(R + \frac{\gamma^2 R^2}{\sigma^2}\Big).$$

Finally we aim for establishing a lower bound on $\sigma^2 d - 2\,\boldsymbol{u}^\top\mathbf{\Sigma}\,\boldsymbol{u}$. This is achieved by the following observation

$$\frac{R^2\lambda}{(\lambda + R)^2} \;\le\; \frac{R^2(\lambda + R)}{(\lambda + R)^2} = \frac{R^2}{\lambda + R} \;\le\; R,$$

Hence

$$\boldsymbol{u}^\top\mathbf{\Sigma}\,\boldsymbol{u} \;=\; \sum_{i=1}^{d}\frac{R^2\lambda_i}{(\lambda_i + R)^2}\,\tilde{w}_i^2 \;\le\; R\sum_{i=1}^{d}\tilde{w}_i^2 = R\,\|\boldsymbol{w}\|^2 = R\,d.$$

Therefore

$$\sigma^2 d - 2\,\boldsymbol{u}^\top\mathbf{\Sigma}\,\boldsymbol{u} \;\ge\; d\,(\sigma^2 - 2R).$$

Putting both the results together, when $R/\sigma^2 \ll 1$ we get

$$\left|\frac{\partial\log\delta}{\partial\log n}\right| \;\le\; \cdot\frac{\frac{8}{27}\,d\,(R + \gamma^2 R^2/\sigma^2)}{d(\sigma^2 - 2R)} \;=\; \frac{8}{27}\cdot\frac{R}{\sigma^2}\cdot\frac{1 + \gamma^2 R/\sigma^2}{1 - 2R/\sigma^2} \;\ll\; 1$$

$\square$

# F   GOING BEYOND INDEPENDENT INFERENCE TIME SAMPLING: SELF-CONSISTENCY AND BEAM SEARCH

In this appendix, we outline two speculative ideas that go beyond independent inference time sampling discussed in this paper. It would be fascinating to explore these directions in the future.

## F.1   SELF-CONSISTENCY

Self-consistency promotes candidates supported by their peers. We can model this by augmenting the selection score with a consensus kernel:

$$s_i = \sum_{j \neq i} K_\tau(y_i, y_j), \qquad K_\tau(y, y') = \exp\left(-\frac{(y - y')^2}{\tau}\right),$$

and scoring each candidate with

$$\underbrace{r(y_i, \mathbf{x})}_{\text{alignment to reward}} + \beta \underbrace{\log s_i}_{\text{alignment to consensus}},$$

for some $\beta \geq 0$. The softmax then uses $\exp\left([r_R(y_i, \mathbf{x}) + \beta \log s_i]/T\right)$. For the purpose of analytical calculations it is reasonable to approximate $s_i \to \mathbb{E}[s_i \mid y_i]$ in this expression. This in effect rescales $T$ and shifts $\mathbf{w}_R$. The theoretical analysis can be performed following similar steps as in the paper.

## F.2   BEAM SEARCH

Fix a search budget $K$ (total proposals explored by the decoder) and a beam width $B \in \{1, \ldots, K\}$ (candidates kept after pruning). A beam search operator $\mathcal{B}_{K,B}$ keeps the $B$ proposals:

$$\{Y_{(1)}, \ldots, Y_{(B)}\} = \mathcal{B}_{K,B}\left(\{Y_i\}_{i=1}^K\right) = \arg\text{top-}B\,\{-|Y_i - m|\}_{i=1}^K.$$

These $B$ kept candidates are then passed to our reward-weighted selector (softmax over $r(y, \mathbf{x})$ at temperature $T$).

The marginal distribution of a kept value under $\mathcal{B}_{K,B}$ is well-approximated by (we take correlations into account later):

$$q_{\text{beam}}(y \mid \mathbf{x}, \mathcal{D}) \propto \mathcal{N}(y \mid m, s^2)\,\mathbf{1}\{|y - m| \leq t_B\},$$

where the truncation threshold $t_B$ is chosen so that the expected keep rate matches $B/K$:

$$\Pr\left(|Y - m| \leq t_B\right) = \frac{B}{K}$$

Under symmetric truncation, the mean remains $m$ and the variance $s^2$ contracts to $s_{\text{beam}}^2$.

Even if $Y_1, \ldots, Y_K$ are i.i.d., the kept vector $(Y_{(1)}, \ldots, Y_{(B)})$ is not independent (order-statistic coupling). A standard correction is to replace $k$ by an effective size $k_{\text{eff}}$ such that $Var(\bar{g}_B) = Var(g(Y_{(i)}))/k_{\text{eff}}$, $\bar{g}_B = \frac{1}{B}\sum_{i=1}^B g(Y_{(i)})$. Simplest case would correspond to choosing $g(x) = x$.

In summary, we expect that the replacements $s^2 \to s_{\text{beam}}^2$, $k \to k_{\text{eff}}$ in our formula would capture the effect of beam search.

# G  ADDITIONAL EXPERIMENTAL RESULTS

In this Appendix we present additional numerical results for a broader domain of parameters compared to the main text. We see that the patterns explained in the main body of the paper is realized in a broad domain of parameters.

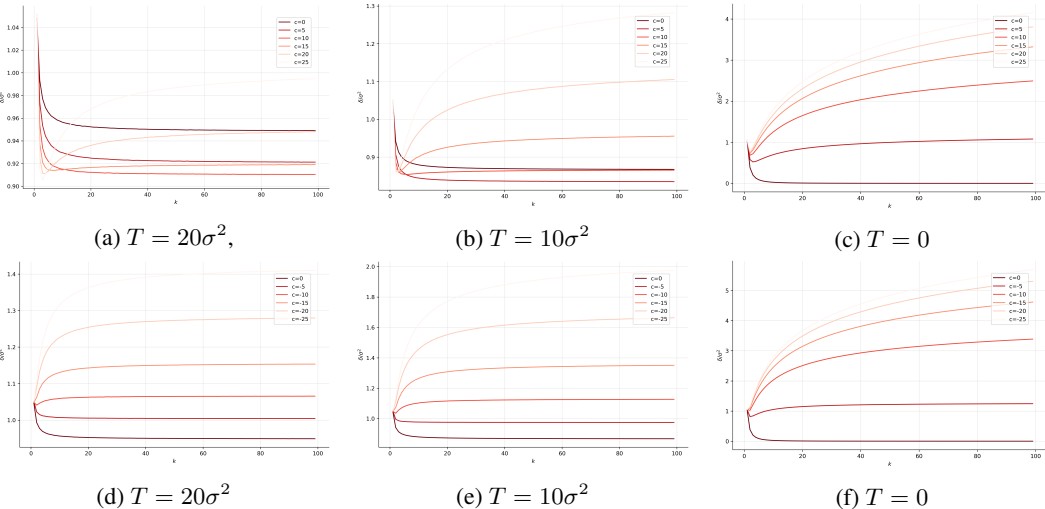

(a) $T = 20\sigma^2$,  (b) $T = 10\sigma^2$  (c) $T = 0$

(d) $T = 20\sigma^2$  (e) $T = 10\sigma^2$  (f) $T = 0$

Figure 6: In the plot we have chosen $S = 1, \sigma = 10^{-4}, \gamma = 10^{-3}, n = 10^4, d = 10^1$ and sampled teacher weight $\mathbf{w}_T \sim \mathcal{N}(0, 2^2\mathbf{I})$. We have parameterized the reward weight as follows: $\mathbf{w}_R = (1 + cR/(R + S^2))\mathbf{w}_T$. The plot shows asymmetry between $c > 0, c < 0$ regions as explained in the main text.

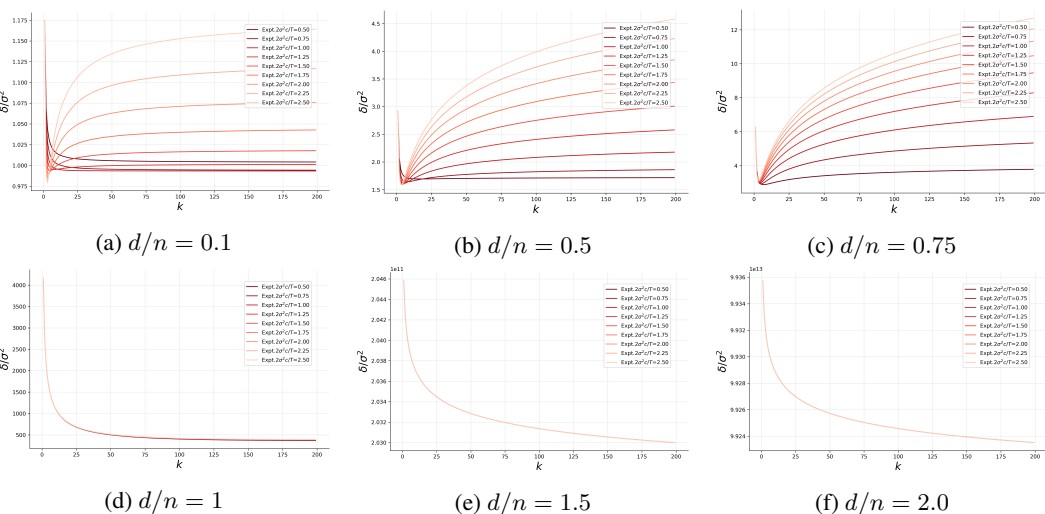

(a) $d/n = 0.1$  (b) $d/n = 0.5$  (c) $d/n = 0.75$

(d) $d/n = 1$  (e) $d/n = 1.5$  (f) $d/n = 2.0$

Figure 7: In the plot we have chosen $S = 1, \sigma = 10^{-4}, \gamma = 10^1, n = 10^2, T = 20\sigma^2$ and sampled teacher weight $\mathbf{w}_T \sim \mathcal{N}(0, 2^2\mathbf{I})$. We have parameterized the reward weight as follows: $\mathbf{w}_R = (1 + cR/(R + S^2))\mathbf{w}_T$. We see that for $d \geq n$ generalization error shows different pattern compared to $d < n$. For $d < n$ we see features that are discussed in the main text. We note that as $d/n$ increases $\delta$ at fixed $k$ generally increases. Nevertheless, even for $d \geq n$, an increase in $k$ decreases $\delta$ for a wide range of $\mathbf{w}_R$.

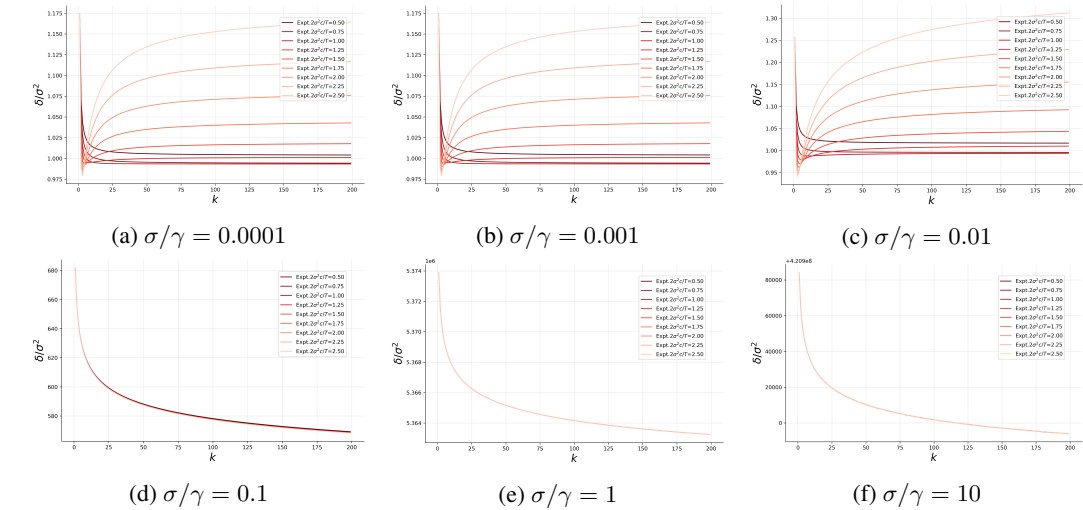

(a) $\sigma/\gamma = 0.0001$   (b) $\sigma/\gamma = 0.001$   (c) $\sigma/\gamma = 0.01$

(d) $\sigma/\gamma = 0.1$   (e) $\sigma/\gamma = 1$   (f) $\sigma/\gamma = 10$

Figure 8: In the plot we have chosen $S = 1, \sigma = 10^{-4}, n = 10^2, d = 10^1, T = 20\sigma^2$ and sampled teacher weight $\mathbf{w}_T \sim \mathcal{N}(0, 2^2\mathbf{I})$. We have parameterized the reward weight as follows: $\mathbf{w}_R = (1 + cR/(R + S^2))\mathbf{w}_T$. We see that for large $\sigma/\gamma$ generalization error shows different pattern compared to small $\sigma/\gamma$. Plots show similarity with the plot of $\delta$ vs $d/n$ - in the language of deterministic equivalence both of these are related to the similar change of the un-renormalized ridge.

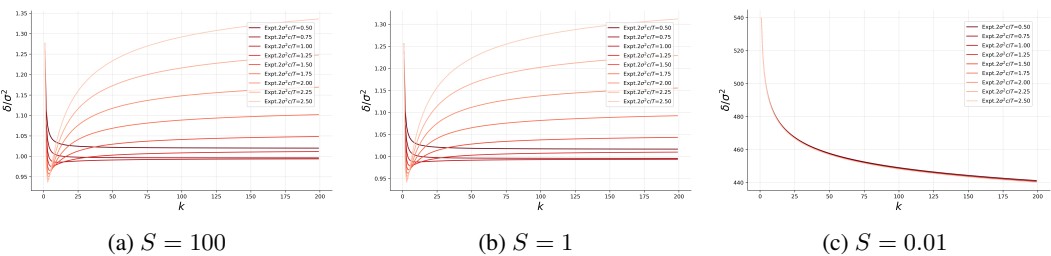

(a) $S = 100$   (b) $S = 1$   (c) $S = 0.01$

Figure 9: In the plot we have chosen $T = 20\sigma^2, \sigma = 10^{-4}, \gamma = 10^{-2}, n = 10^2, d = 10^1$ and sampled teacher weight $\mathbf{w}_T \sim \mathcal{N}(0, 2^2\mathbf{I})$. We have parameterized the reward weight as follows: $\mathbf{w}_R = (1 + cR/(R + S^2))\mathbf{w}_T$. The plot shows as $S$ is lowered beyond a critical value we see a sharp change of features.

