# OpenReview forum: "A solvable model of inference-time scaling"
_ICLR.cc/2026/Conference — Submitted to ICLR 2026_

### Official Review · Reviewer_QdH3 · 2025-10-27

**Soundness:** 2
**Presentation:** 3
**Contribution:** 2
**Rating:** 2
**Confidence:** 4

**Summary:**

The paper proposes a Bayesian linear regression model to study the performance of best-of-k sampling in inference-time scaling. The model involves elements such as teaching model, reward modeling, and sampling temperature. The theoretical results provide insights into the relationship between the model performance with (i) the temperature, (ii) the number of samples k, and (iii) the goodness of the reward model.

**Strengths:**

The paper is well-written, and the theoretical analysis is thorough and accompanied with discussions and intuitions.

In particular,
- The analysis captures the key factors in inference scaling: temperature, k, and reward model.
- The resulting curves (monotonic and non-monotonic) are both observed in practice. So, to some extent, the implications of the theoretical frameworks match the empirical observations.

**Weaknesses:**

My main concern is whether the model provides insights into the real practical usage of the inference-time scaling.
- As I mentioned above, the resulting curves derived from the theoretical framework match empirical observations in practice. However, the insights from the paper on the choices of the parameters, such as k and temperature, provides no guidance on their practical choice. The actual behavior of the inference-time scaling might not be able to be captured by a linear regression setup.
- There is no numerical experiment on running inference-time scaling for real LLM models in this paper, which makes the results less convincing. In this light, the paper is more like a thorough analysis of a (Bayesian) linear regression model but its technical contribution doesn't go beyond that.

From the modeling viewpoint:
- Can the current framework capture techniques such as beam-search-based generation and the self-consistency approach in inference-time scaling?

**Questions:**

See above weaknesses. Also, I think the authors should think more about how to make the framework more realistic to capture the real usage of inference-time scaling, or, how to convince the practioners using inference-time scaling that the framework can guide their daily practice.

---

> ### Author Response · Authors · 2025-11-25
>
> We thank the reviewer for the thoughtful assessment and for highlighting our theoretical contributions.  We added a new section, called ``Qualitative agreement with large language model reasoning'', please take a look.  We respond to the main concerns and summarize concrete revisions made in the revised manuscript.
>
> - The section presents real LLM experiments validating key qualitative predictions of our theory. In the new section, we evaluate Llama-3-8B-Instruct on openai/gsm8k with $k\in\{1,4,8,16,32,64,128\}$ candidate generations per question (8 chain-of-thought demonstrations in the prompt). Each candidate is scored by a separate judge model Mistral-7B-Instruct-v0.3 to produce rewards $r(y_i, x)$. We then perform reward-weighted selection via softmax at temperature $T$, and report the generalization proxy already formalized in the paper,
> $$\delta=-E_{x}Σ_{i=1}^{k}\frac{e^{r(y_i,x)/T}}{\sum_{j=1}^{k}e^{r(y_j,x)/T}}v(y_i,x)$$
> where $v(y_i,x)\in\{0,1\}$ denotes the correctness, it is $1$ when the response is correct and $0$ otherwise. We show that for fixed $k=64$, $δ$ is U-shaped with a clear optimum in $T$ matching results for the linear model. We also present an experiment at fixed temperature $T=10$ showing that $δ$ as a function of $k$ has a global minimum qualitatively matching theoretical discussions.  A similar fact has been observed in large language models before in Snell et al 2025 and Chen et al 2024. If the paper is accepted for publication in camera camera-ready version, we can add more detailed experimental results.
>
> - We also outline how our observation about the optimal reward differing from the teacher can be implemented in LLMs.
>
> - We would also emphasize that our paper is the first solvable model that gives an opportunity to discuss training and test time compute trade-offs. At present, we lack any theoretical understanding of inference time scaling; therefore, we request the reviewer to consider that our paper serves as a concrete progress in this direction.
>
> - ``Can the current framework capture techniques such as beam-search-based generation and the self-consistency approach in inference-time scaling?''
> We thank the reviewer especially for this question. We added a new appendix F called Going beyond independent inference time sampling: self-consistency and beam search to discuss future directions addressing this. In short, both self-consistency and beam-search can be taken into account in an analytically solvable manner. Please take a look at the appendix. However complete analysis of this is beyond the scope of the current paper.
>
> If you find our response to your concerns satisfactory, please consider increasing the review score.

---

### Official Review · Reviewer_TCYV · 2025-11-01

**Soundness:** 2
**Presentation:** 3
**Contribution:** 2
**Rating:** 4
**Confidence:** 3

**Summary:**

This paper introduces an analytically solvable model to theoretically investigate the principles of inference-time scaling. The authors model this problem using Bayesian linear regression in a high-dimensional, teacher-student framework. At inference time, k candidate predictions are sampled and then selected using a softmax function over a quadratic reward, controlled by a temperature parameter T. The paper derives closed-form expressions for the generalization error and analyzes its dependence on the number of samples (k), temperature (T), and the quality of the reward model (i.e., its alignment with the true data-generating "teacher" model).

**Strengths:**

1. The community has observed many empirical phenomena about inference-time compute (e.g., "best-of-k", "self-consistency"), but a clear theoretical understanding is lacking. This paper fills that gap

2. Despite its simplicity, the model successfully reproduces several non-trivial behaviors seen in massive, complex models like LLMs. This paper provides a simple, intuitive reason: an imperfect reward model will eventually favor samples that are "good" according to its flawed criteria but bad according to the true objective, and more samples increase the chance of finding such a "trap" sample.

3. The theoretical derivations (based on high-dimensional statistics and deterministic equivalents) are thoroughly validated against numerical simulations of the model itself (e.g., Figures 2 and 4). The extremely close match between the "D.E." (theory) and "Expt." (simulation) lines gives high confidence that the mathematical analysis is correct.

**Weaknesses:**

1. The model is based on linear regression, whereas modern applications use highly non-linear Transformer architectures.

2. The reward is a simple quadratic function. Real-world reward models (often used for RLHF) are complex neural networks trained to predict human preferences. The data is assumed to be Gaussian, which is very different from the structured, discrete nature of language.
This gap means the specific quantitative results (e.g., the exact formula for the 1/k² decay) may not transfer directly to LLMs. However, the qualitative insights and intuitions derived are still extremely valuable.

3. This paper does not contain experiments on real-world datasets or with actual LLMs. The validation is purely "internal" (checking the theory against simulations of the same theoretical model). While this is standard for a purely theoretical paper, it leaves the question of how well these insights generalize to practice open. An ideal follow-up work would be to test if the principles derived here (e.g., the relationship between reward model quality and optimal k) hold up in experiments with a real LLM.

**Questions:**

n/a

---

> ### Author Response · Authors · 2025-11-25
>
> We thank the reviewer for the thoughtful assessment and for highlighting our theoretical contributions.  We added a new section, called ``Qualitative agreement with large language model reasoning'', please take a look.  We respond to the main concerns and summarize concrete revisions made in the revised manuscript.
>
> - The section presents real LLM experiments validating key qualitative predictions of our theory. In the new section, we evaluate Llama-3-8B-Instruct on openai/gsm8k with $k\in\{1,4,8,16,32,64,128\}$ candidate generations per question (8 chain-of-thought demonstrations in the prompt). Each candidate is scored by a separate judge model Mistral-7B-Instruct-v0.3 to produce rewards $r(y_i, x)$. We then perform reward-weighted selection via softmax at temperature $T$, and report the generalization proxy already formalized in the paper,
> $$\delta=-E_{x}Σ_{i=1}^{k}\frac{e^{r(y_i,x)/T}}{\sum_{j=1}^{k}e^{r(y_j,x)/T}}v(y_i,x)$$
> where $v(y_i,x)\in\{0,1\}$ denotes the correctness, it is $1$ when the response is correct and $0$ otherwise. We show that for fixed $k=64$, $δ$ is U-shaped with a clear optimum in $T$ matching results for the linear model. We also present an experiment at fixed temperature $T=10$ showing that $δ$ as a function of $k$ has a global minimum qualitatively matching theoretical discussions.  A similar fact has been observed in large language models before in Snell et al 2025 and Chen et al 2024. If the paper is accepted for publication in camera camera-ready version, we can add more detailed experimental results.
>
> - We also outline how our observation about the optimal reward differing from the teacher can be implemented in LLMs.
>
> - We would also emphasize that our paper is the first solvable model that gives an opportunity to discuss training and test time compute trade-offs. At present, we lack any theoretical understanding of inference time scaling; therefore, we request the reviewer to consider that our paper serves as a concrete progress in this direction.
>
> In addition, we added a new appendix F called ``Going beyond independent inference time sampling: self-consistency and beam search'' to discuss future directions.
>
> If you find our response to your concerns satisfactory, please consider increasing the review score.

---

### Official Review · Reviewer_hztQ · 2025-11-03

**Soundness:** 2
**Presentation:** 2
**Contribution:** 2
**Rating:** 4
**Confidence:** 2

**Summary:**

This paper introduces an analytically tractable model of inference-time scaling using Bayesian linear regression with reward-weighted sampling, deriving closed-form expressions for generalization error in the high-dimensional limit. The authors prove that when the reward model is well-aligned with the teacher, error decreases monotonically with inference samples $k$ (scaling as $\Theta(1/k^2)$ in the best-of-k limit), but substantial reward misspecification induces a finite optimal $k$ and optimal temperature. The theory delineates parameter regimes where scaling inference-time compute is provably more effective than collecting additional training data, though this advantage degrades as task difficulty increases.

**Strengths:**

1. The model provides closed-form solutions for generalization error that can be directly computed and verified, unlike most existing work that relies purely on empirical observations.

2. The paper identifies concrete conditions (optimal temperature, optimal k, reward alignment thresholds) that practitioners can actually use when designing inference-time systems.

3. The theoretical framework quantifies when to invest compute in inference versus training, addressing a key resource allocation question that lacks prior rigorous analysis.

**Weaknesses:**

1. Oversimplified model: The paper only studies linear regression with quadratic rewards and Gaussian assumptions, while real LLMs involve highly nonlinear neural networks, complex reward models, and non-Gaussian data distributions.

2. No validation on real LLMs: All experiments use synthetic linear regression data, and the theoretical insights (optimal temperature, optimal k) are not verified on actual language models, with connections to LLM phenomena relying mainly on citations rather than direct evidence.

3. The paper focuses on best-of-k and reward-weighted sampling but does not provide theoretical analysis for majority voting or meta-voter aggregation schemes, which are commonly used in practice.

The paper's core contribution is providing an analytically tractable toy model, but it remains far from explaining inference-time compute behavior in real LLMs. It serves more as a proof of concept, demonstrating that certain phenomena (e.g., non-monotonic k, optimal temperature) can be theoretically understood in simplified settings, but significant follow-up work is needed to bridge the gap between the theoretical model and practical systems before it can truly guide real-world applications.

**Questions:**

Please see above.

---

> ### Author Response · Authors · 2025-11-25
>
> We thank the reviewer for the thoughtful assessment and for highlighting our theoretical contributions.  We added a new section, called ``Qualitative agreement with large language model reasoning'', please take a look.  We respond to the main concerns and summarize concrete revisions made in the revised manuscript.
>
> - The section presents real LLM experiments validating key qualitative predictions of our theory. In the new section, we evaluate Llama-3-8B-Instruct on openai/gsm8k with $k\in\{1,4,8,16,32,64,128\}$ candidate generations per question (8 chain-of-thought demonstrations in the prompt). Each candidate is scored by a separate judge model Mistral-7B-Instruct-v0.3 to produce rewards $r(y_i, x)$. We then perform reward-weighted selection via softmax at temperature $T$, and report the generalization proxy already formalized in the paper,
> $$\delta=-E_{x}Σ_{i=1}^{k}\frac{e^{r(y_i,x)/T}}{\sum_{j=1}^{k}e^{r(y_j,x)/T}}v(y_i,x)$$
> where $v(y_i,x)\in\{0,1\}$ denotes the correctness, it is $1$ when the response is correct and $0$ otherwise. We show that for fixed $k=64$, $δ$ is U-shaped with a clear optimum in $T$ matching results for the linear model. We also present an experiment at fixed temperature $T=10$ showing that $δ$ as a function of $k$ has a global minimum qualitatively matching theoretical discussions.  A similar fact has been observed in large language models before in Snell et al 2025 and Chen et al 2024. If the paper is accepted for publication in camera camera-ready version, we can add more detailed experimental results.
>
> - We also outline how our observation about the optimal reward differing from the teacher can be implemented in LLMs.
>
> - We would also emphasize that our paper is the first solvable model that gives an opportunity to discuss training and test time compute trade-offs. At present, we lack any theoretical understanding of inference time scaling; therefore, we request the reviewer to consider that our paper serves as a concrete progress in this direction.
>
> In addition, we added a new appendix F called ``Going beyond independent inference time sampling: self-consistency and beam search'' to discuss future directions.
>
> If you find our response to your concerns satisfactory, please consider increasing the review score.

---

### Official Review · Reviewer_k4BJ · 2025-11-09

**Soundness:** 2
**Presentation:** 2
**Contribution:** 2
**Rating:** 2
**Confidence:** 3

**Summary:**

The paper proposes a solvable model of inference-time scaling based on Bayesian linear regression with reward-weighted sampling, deriving analytic expressions for generalization error under different temperatures, reward alignments, and sample counts. The analysis connects these behaviors to patterns reported in recent LLM work.

**Strengths:**

- Clean and rigorous theoretical development.
- The model is mathematically elegant and yields interpretable predictions (e.g., optimal k and temperatures).
- The paper provides useful intuition about how reward quality influences inference-time compute.

**Weaknesses:**

- **No experiments on any real model.** All empirical results come from the same synthetic linear teacher–student setup used in the derivations. There is no validation on actual neural networks or LLM inference-time sampling. As a result, none of the claims about LLM behaviors are verifiable, and the practical relevance of the theory remains untested. This is the major flaw of this paper.

- The experimental section is minimal and does not explore settings beyond the analytic assumptions. The paper seems overstate the connection between this toy model and real LLM inference dynamics.

**Questions:**

NA

---

> ### Author Response · Authors · 2025-11-25
>
> We thank the reviewer for the thoughtful assessment and for highlighting our theoretical contributions.  We added a new section, called ``Qualitative agreement with large language model reasoning'', please take a look.  We respond to the main concerns and summarize concrete revisions made in the revised manuscript.
>
> - The section presents real LLM experiments validating key qualitative predictions of our theory. In the new section, we evaluate Llama-3-8B-Instruct on openai/gsm8k with $k\in\{1,4,8,16,32,64,128\}$ candidate generations per question (8 chain-of-thought demonstrations in the prompt). Each candidate is scored by a separate judge model Mistral-7B-Instruct-v0.3 to produce rewards $r(y_i, x)$. We then perform reward-weighted selection via softmax at temperature $T$, and report the generalization proxy already formalized in the paper,
> $$\delta=-E_{x}Σ_{i=1}^{k}\frac{e^{r(y_i,x)/T}}{\sum_{j=1}^{k}e^{r(y_j,x)/T}}v(y_i,x)$$
> where $v(y_i,x)\in\{0,1\}$ denotes the correctness, it is $1$ when the response is correct and $0$ otherwise. We show that for fixed $k=64$, $δ$ is U-shaped with a clear optimum in $T$ matching results for the linear model. We also present an experiment at fixed temperature $T=10$ showing that $δ$ as a function of $k$ has a global minimum qualitatively matching theoretical discussions.  A similar fact has been observed in large language models before in Snell et al 2025 and Chen et al 2024. If the paper is accepted for publication in camera camera-ready version, we can add more detailed experimental results.
>
> - We also outline how our observation about the optimal reward differing from the teacher can be implemented in LLMs.
>
> - We would also emphasize that our paper is the first solvable model that gives an opportunity to discuss training and test time compute trade-offs. At present, we lack any theoretical understanding of inference time scaling; therefore, we request the reviewer to consider that our paper serves as a concrete progress in this direction.
>
> In addition, we added a new appendix F called ``Going beyond independent inference time sampling: self-consistency and beam search'' to discuss future directions.
>
> If you find our response to your concerns satisfactory, please consider increasing the review score.

---

### Meta-Review · Area_Chair_9CNW · 2026-01-12

**Summary:**

This submission derives an inference-time scaling law in the context of Bayesian linear regression with Softmax sample selection. All reviewers raised the following concerns.

* The theoretical setting is oversimplified (Bayesian linear regression, quadratic reward, best-of-$N$, etc.), and the practical guidance from the theoretical results is not discussed.

* The paper does not include any LLM experiments to validate the theoretical insights. The authors partly addressed this concern by adding an experiment on best-of-$N$ inference with an LLM-as-judge.

Overall, while the authors addressed some of the reviewers’ concerns, the area chair believes that this submission is currently below the acceptance bar and encourages the authors to improve the manuscript and resubmit to a future venue.

**Reviewer Concerns:**

See above.

**Reviewer Scores:**

All reviewers provided a negative evaluation due to the above concerns, which are not fully addressed by the limited experiments added in the revision.

---

### Decision · Program_Chairs · 2026-01-26

Reject